# Extracellular vesicles from mature dendritic cells (DC) differentiate monocytes into immature DC

Stefan Schierer[1],*, Christian Ostalecki[1],*, Elisabeth Zinser[2], Ricarda Lamprecht[1], Bianca Plosnita[3], Lena Stich[2], Jan Dörrie[1], Manfred B Lutz[4], Gerold Schuler[1], Andreas S Baur[1]

During inflammation, murine and human monocytes can develop into dendritic cells (DC), but this process is not entirely understood. Here, we demonstrate that extracellular vesicles (EV) secreted by mature human DC (maDC) differentiate peripheral monocytes into immature DC, expressing a unique marker pattern, including 6-sulfo LacNAc (slan), Zbtb46, CD64, and CD14. While EV from both maDC and immature DC differentiated monocytes similar to GM-CSF/IL-4 stimulation, only maDC-EV produced precursors, which upon maturation stimulus developed into T-cell–activating and IL-12p70–secreting maDC. Mechanistically, maDC-EV induced cell signaling through GM-CSF, which was abundant in EV as were IL-4 and other cytokines and chemokines. When injected into the mouse skin, murine maDC-EV attracted immune cells including monocytes that developed activation markers typical for inflammatory cells. Skin-injected EV also reached lymph nodes, causing a similar immune cell infiltration. We conclude that DC-derived EV likely serve to perpetuate an immune reaction and may contribute to chronic inflammation.

## Introduction

Numerous functions have been attributed to extracellular vesicles (EV), owing to their rich content of mRNA/miRNA (Valadi et al, 2007; Skog et al, 2008), surface receptors (Thery et al, 2009), ADAM protease enzymatic activity (Lee et al, 2013), and cytokines, chemokines, and other soluble factors (hereafter referred to as CCF) (Lee et al, 2016). Dendritic cell (DC)–derived EV were analyzed early on, following the discovery that MHC class-II–enriched vesicles are able to induce antigen-specific T-cell responses (Raposo et al, 1996). These studies suggested that DC-derived EV have multiple functions in immune regulation and can modulate T-cell responses by interacting with DC as well as T cells (Thery et al, 2002,

2009). In addition to their antigen-presenting capabilities, DC-derived EV were found to activate NK cells and, through the presence of TNF, FasL, and TRAIL, kill tumor cells (Zitvogel et al, 1998; Tel et al, 2014).

Besides conventional or classical hematopoietic stem cell–derived DC (cDC1/cDC2) and plasmacytoid DC (pDC) (Wu & Liu, 2007; Liu et al, 2009; Mildner & Jung, 2014), monocyte-derived cell populations exert antigen-presenting immune functions (Geppert & Lipsky, 1989; Cros et al, 2010; Schlitzer et al, 2015; Jakubzick et al, 2017; Lutz et al, 2017). In addition, there is a growing consensus that in humans monocytes give rise to inflammatory DC [(inf)DC] (Leon et al, 2007; Shortman & Naik, 2007; Segura et al, 2013), which are characterized by the expression of a set of markers (HLA-DR, CD11c, BDCA1, CD1a, FcεRI, CD206, CD172a, CD14, CD11b, and Zbtb46) (Segura & Amigorena, 2013). While these cells are functionally similar to conventional DC, their transcriptome is distinct (Briseno et al, 2016; Sander et al, 2017) and they are considered as inflammatory monocyte-derived cells (Schlitzer et al, 2015). Hence, they may occupy the far end of a versatile and condition-adaptable monocyte cell population. The previously described human 6-sulfo LacNAc (slan)–expressing DC (Schakel et al, 2002), which are thought to originate from blood CD16[+] monocytes, are also classified into this inflammatory cell type (Segura et al, 2013; Schlitzer et al, 2015). In line with this conclusion, slanDC were detected in tissue of chronic inflammatory diseases including psoriasis and lupus erythematodes (Hansel et al, 2011, 2013).

Like in humans, there is an evolving consensus that in mice inflammatory DC develop from monocytes (Segura et al, 2013). This seems to occur particularly in an inflamed environment, as demonstrated in different mouse inflammation models (Naik et al, 2006; Hohl et al, 2009; Greter et al, 2012). Supporting this conclusion, the number of developing (inf)DC is not affected in Flt3L–/– mice (Plantinga et al, 2013), while cDC, which depend on Flt3L stimulation, are dramatically reduced (Waskow et al, 2008). Conversely, in mice deleted for the monocyte migration marker CCR2, (inf)DC are greatly

[1]Department of Dermatology, University Hospital Erlangen, Kussmaul Campus, Erlangen, Germany   [2]Department of Immune Modulation, University Hospital Erlangen, Kussmaul Campus, Erlangen, Germany   [3]TissueGnostics GmbH, Wien, Austria   [4]Institute of Virology and Immunobiology, Würzburg, Germany

Correspondence: andreas.baur@uk-erlangen.de
*Stefan Schierer and Christian Ostalecki contributed equally to this work.

diminished in inflamed tissue (Naik et al, 2006; Osterholzer et al, 2009).

Mouse monocyte–derived (*inf*)DC were initially identified as being positive for MHC-II, CD11b, CD11c, F4-80, and Ly6C (Leon et al, 2007), and additional markers have since been identified, including FcεRI and CD64 (Plantinga et al, 2013). However, the immediate precursors of these cells are not sufficiently characterized, and it is not clear how these monocyte-derived DC precursors develop. Conventional wisdom, as well as the rich work done so far, would suggest that this depends not only on the inflammatory cytokine milieu but also on

additional parameters including tissue- and potentially host-specific factors (Hohl et al, 2009; Alcantara-Hernandez et al, 2017).

In our previous work, we had noticed that monocytes efficiently ingest EV (Lee et al, 2013). Here, we demonstrate that DC-derived EV, due to their rich content of cytokines, chemokines, and soluble ligands, have the capacity to mobilize and differentiate monocytes into a variety of phenotypes, likely to amplify and adapt an inflammatory immune response to a given situation. Our study points at the importance of EV for the function and activation of the immune cell network.

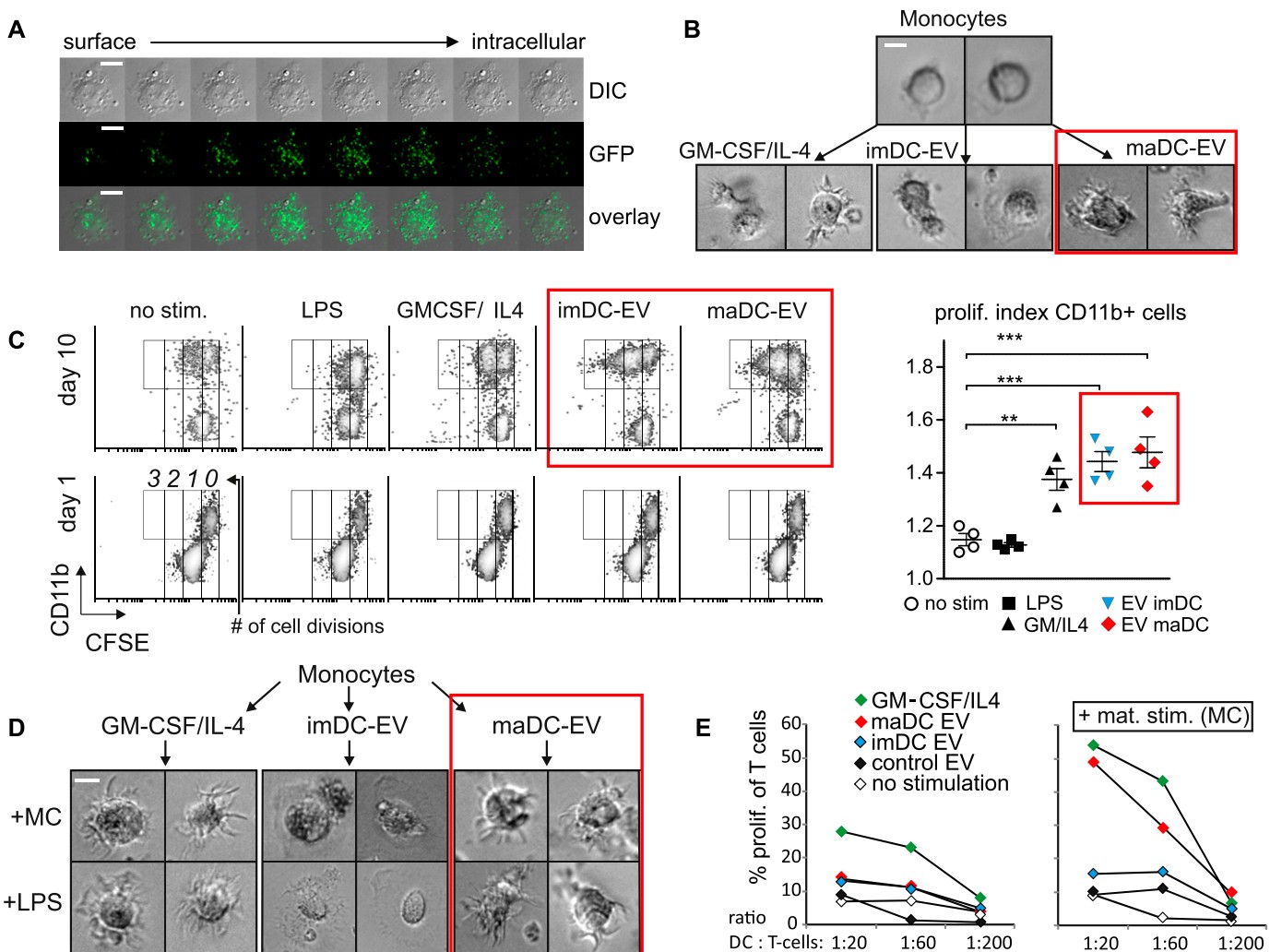

**Figure 1. DC-derived EV differentiate monocytes.**
**(A)** Monocytes ingest EV. Monocytes were incubated with DC-derived and GFP-labeled EV for 3 h, washed, and subsequently analyzed by confocal microscopy using Z-stack imaging. DIC: differential interference contrast. **(B)** maDC-EV differentiate a DC-like morphology in monocytes. Monocytes were incubated with the dose of EV (30 μg for $10^6$ cells) derived from imDC and maDC or stimulated with GM-CSF/IL-4 for 6 d. Subsequently, images were taken from representative cells. **(C)** DC-EV induce proliferation in monocytes. PBMCs were labeled with CFSE and treated either with a single dose of imDC or maDC-derived EV (50 μg) or cytokines or LPS as indicated. CFSE dilution in CD11b[+] cells was determined at day 1 and day 10 by flow cytometry, and the number of cell divisions is indicated. The graph summarizes the results from four different donors; one representative result is shown on the left side of the panel. Results are presented as mean ± SEM; statistical significance was analyzed by one-way ANOVA: *P < 0.05, **P < 0.01, and ***P < 0.005. **(D)** maDC-EV–treated monocytes maintain a DC-like morphology upon exposure to maturation stimuli. Same experimental setup as in (B). Subsequently, cells were incubated for 24 h with a MC (IL-1β, IL-6, TNF-α, and PGE₂) or LPS and images were taken from representative cells. **(E)** maDC-EV–treated monocytes that received a maturation stimulus induce T-cell proliferation. Monocytes incubated with imDC and maDC-derived EV (10 μg), or stimulated with GM-CSF/IL-4 (serving as positive control) for 6 d, either received a maturation stimulus (MC) or were left untreated. Subsequently, CFSE-labeled T cells were co-incubated at a defined ratio as indicated and proliferation of cells was determined by radiolabeled thymidine incorporation. Shown is one representative experiment of five performed with different donors (see Fig S3A). Scale bars represent 7.5 μm.

# Results

### DC-derived EV differentiate monocytes

Immature DC (imDC) and mature DC (maDC) were shown to produce substantial amounts of EV (Zitvogel et al, 1998; Thery et al, 2009). We speculated that monocytes might be the physiological target cells for these DC-derived EV. To substantiate this assumption, we generated labeled EV by electroporating GFP RNA into human monocyte-derived maDC as described previously from our institution (Gerer et al, 2017). The vesicles were purified by differential centrifugation and incubated with PBMCs for 3 h, which were subsequently analyzed by flow cytometry. Only 16% of the lymphocytes, but 70% of the monocytes, gave a positive signal for GFP (Fig S1). To confirm that the EV were ingested, we analyzed GFP-positive monocytes by confocal microscopy. This revealed an intracellular speckled distribution of the GFP signal as expected for the uptake of vesicular structures (Fig 1A).

To determine whether the EV uptake had target cell effects, we incubated primary monocytes with a single dose of an EV preparation (10 $\mu$g for 2.5 × 10$^5$ cells), obtained from monocyte-derived imDC and maDC. The latter were generated by the standard protocol using a cytokine maturation cocktail (MC: IL-1$\beta$, IL-6, TNF, and PGE$_2$) (Jonuleit et al, 1997). This 10-$\mu$g stimulus contained EV produced by 4 × 10$^6$ imDC or maDC in 24 h (see the Materials and Methods section for details). The concentration of CCF in 10 $\mu$g of maDC-derived EV (maDC-EV) is listed in Fig S2A.

We also determined the ratio of EV-associated and non-EV–associated CCF in DC supernatants (Fig S2B). This revealed that only a fraction of each CCF was secreted through EV; however, this seemed to depend on the factor, as, for example, the EV concentration of IFN-$\gamma$ was 40-fold less in EV as compared with the supernatant, whereas IL-21 was more than 1,000-fold less in EV.

After 6 d of culture with 10-$\mu$g EV preparations, the cells were first examined by light microscopy. Predominantly, maDC-EV induced morphological changes that were typical for DC including the characteristic veils emanating from the plasma membrane (Fig 1B, red box). The cells increased in number and appeared to become larger (see also Fig S3A), a sign of maturation (Jakubzick et al, 2017).

To substantiate the impression of maturing and proliferating cells, PBMCs were labeled with CFSE and stimulated with imDC-EV, maDC-EV (1 dose of 10 $\mu$g each), GM-CSF/IL-4, or LPS or left untreated. Indeed, the EV-treated CD11b$^+$ fraction (Fig 1C, red boxes), which includes monocytes and DC, showed an increased proliferation index, which, in comparison with non-stimulated cells, was more pronounced than seen with GM-CSF/IL-4–stimulated cells.

When EV-treated monocytes, after 6 d, were additionally incubated with a DC maturation stimulus (LPS or MC), the maDC-EV–incubated cells maintained their DC-like morphology (Fig 1D, red box) and were indistinguishable from monocyte-derived maDC generated by standard in vitro stimulation. Conversely, cells treated with imDC-EV did not differentiate in this fashion and adopted a more macrophage-like appearance (Fig 1D).

We then examined the immunostimulatory potential of these cells in a standard MLR. Without a classical maturation stimulus, the EV-treated monocytes had a low stimulatory potential that was exceeded by GM-CSF/IL-4–generated imDC (Fig 1E, left panel, and Fig S3A). However, after these cells were stimulated with a MC, standard monocyte-derived imDC (serving as positive control) and maDC-EV–treated monocytes showed an increased potential to induce T-cell proliferation. Conversely, imDC-EV–treated or untreated monocytes failed to do so (Fig 1E, right panel, and Fig S3A). These morphological and functional findings suggested that maDC-EV differentiated monocytes into cells resembling or constituting immature monocyte-derived DC.

To generate standard monocyte-derived imDC, 423 ng/ml GM-CSF (3 × 800 units) is needed (Jonuleit et al, 1997). We measured around 2.5-ng GM-CSF in 10-$\mu$g maDC-EV preparation (Fig S2A). Hence, there was at least 170-fold more soluble GM-CSF needed to differentiate monocytes in this fashion as compared with EV-associated GM-CSF.

### maDC-EV induce a marker pattern on monocytes similar to that seen on monocyte-derived DC

To get more insight into this differentiation process, we analyzed these cells for typical myeloid cell and DC surface markers by flow cytometry (Fig 2A). imDC-/maDC-EV–incubated peripheral monocytes developed a marker pattern that was clearly distinct from untreated monocytes and similar to patterns found on GM-CSF/IL-4–stimulated monocytes; however, significant differences were also observed. For example, maDC-EV–treated monocytes maintained CD14 and CD163 expression, up-regulated the monocyte-typical marker CD64, and harbored low or lower levels of CD1d, CD1c, and CD209 (Fig 2A; representative graphs in Fig S3B). Overall, however, these cells expressed DC-typical surface markers, including CD86, CD70, CD11c, CD40, and Zbtb46 (red box). The latter was reported to distinguish DC from other immune cells including monocytes (Meredith et al, 2012; Satpathy et al, 2012). Surprisingly, 6-sulfo LacNAc (slan) was also detected, the name-giving marker of the previously described inflammatory and monocyte-derived slanDC (Fig 2A, blue box) (Schakel et al, 2002). Other markers were similarly up-regulated in GM-CSF/IL-4– and EV–treated monocytes, including CD206, CD11b, HLA-DR, CD172, and CD205, whereas FC$\varepsilon$R1 and CD192 changed their expression minimally.

One of the key features of maDC, including inflammatory slanDC, is their capacity to secrete IL-12p70 (Macatonia et al, 1995; Heufler et al, 1996; Schakel et al, 2006). We asked whether maDC-EV–treated monocytes could be stimulated/matured to secrete IL-12. Monocytes were treated for 6 d with imDC-EV, maDC-EV, or GM-CSF/IL-4 and subsequently stimulated for 6 h with the TLR7/-8 agonist R848, before cytokine production was assessed by multiplex technology (BioLegend). While the agonist strongly induced TNF, IL-6, and IL-8 secretion in all conditions, only maDC-EV–treated and GM-CSF/IL-4–stimulated monocytes released IL-12p70 (Fig 2B, red box). Conversely, up-regulation of IL-10 and IL-1$\beta$ was minimal or absent in EV-treated cells as well as GM-CSF/IL-4–treated cells.

In the same cells, a further up-regulation of classical DC maturation markers (CD40, CD80, CD83, and CD86) was observed in both imDC-EV– and maDC-EV–treated cells as assessed by FACS analysis (Fig 2C). In aggregate, these data suggested that maDC-derived EV could differentiate monocytes into cells that were functionally imDC, or EV-induced imDC.

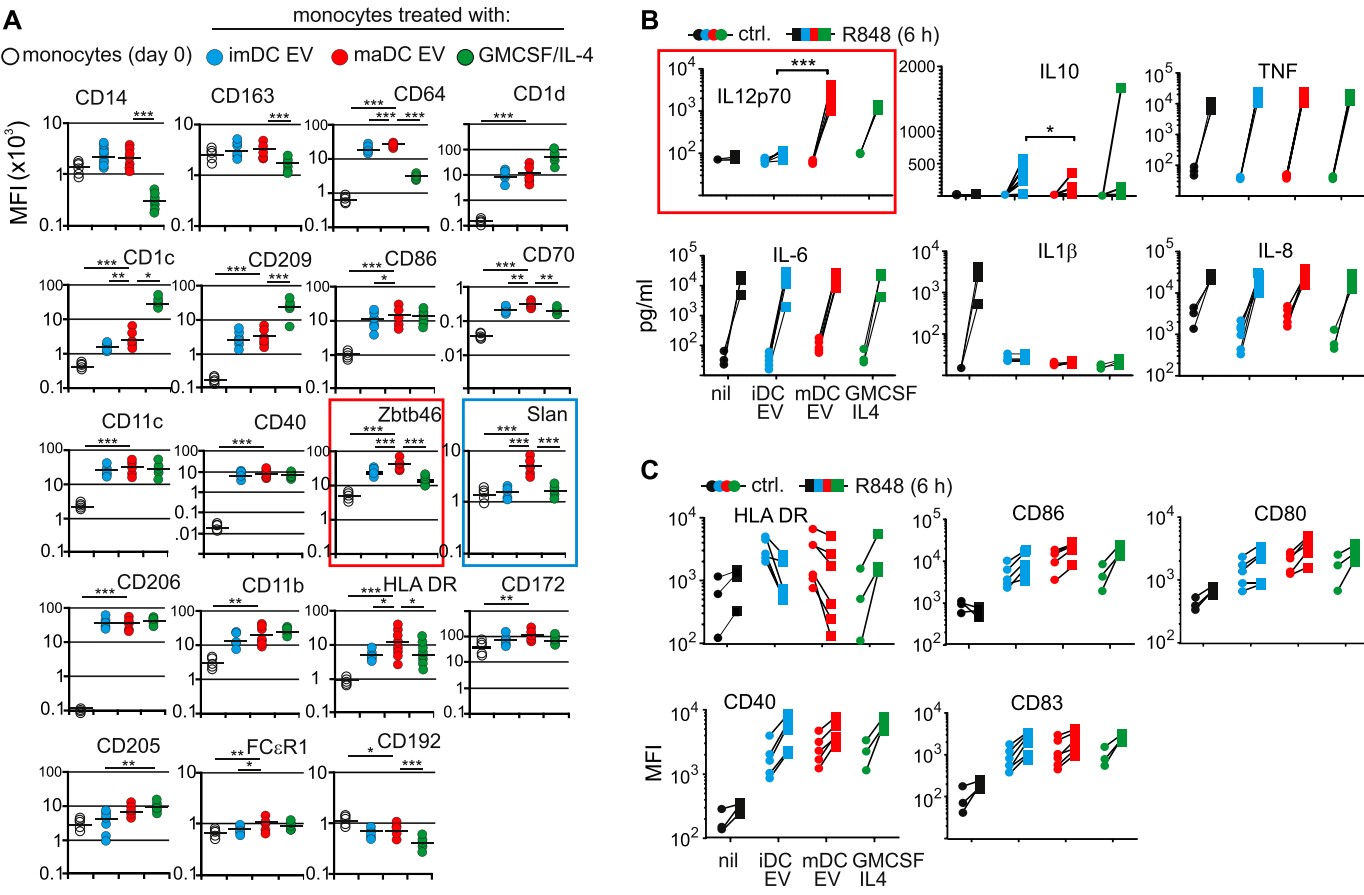

**Figure 2. maDC-EV–treated monocytes develop DC-typical marker expression and factor secretion.**
**(A)** DC-derived EV induce DC-typical marker proteins on monocytes. Peripheral monocytes (2 × 10⁵) were incubated with EV (10 µg) derived from imDC and maDC, or stimulated with GM-CSF/IL-4 for 6 d. Subsequently, cells were analyzed by FACS for the indicated markers. Horizontal black bars represent mean values of all analyzed individual donors. Statistical significance was determined by one-way ANOVA: *P < 0.05, **P < 0.01, and ***P < 0.005. **(B)** maDC-EV–treated monocytes receiving a DC maturation stimulus secrete IL-12p70. Same experimental setup as in (A). The resulting cells were treated with R848 for 6 h or left untreated. Subsequently, the indicated cytokines were measured in the cell culture supernatant. **(C)** Monocytes receiving DC-EV and a DC maturation stimulus express surface markers typical for maDC. Same experimental setup as in (B), and subsequent analysis of indicated surface markers by FACS. In all plots of the figure, each symbol represents one individual donor. The experiments in (A–C) were repeated with different donors, indicated by individual data points.

## maDC-EV convey a cornucopia of effector molecules

A key cytokine in the differentiation of CD14⁺ monocytes towards DC is GM-CSF (Lutz et al, 2017). We examined GM-CSF signaling in EV-treated monocytes by assessing intracellular Stat5 tyrosine phosphorylation through flow cytometry. Monocytes treated with imDC-EV, maDC-EV, or recombinant GM-CSF revealed a strong Stat5 tyrosine phosphorylation as compared with non-stimulated cells (Fig 3A). This effect could be blocked in the presence of an anti-GM-CSF antibody (Fig 3B, red box). In addition, the formation of large CD11b⁺ cells (determined by flow cytometry) was significantly reduced (Fig S4A, red box). Supporting these findings, we could detect GM-CSF on the surface of DC-derived EV by FACS, after the vesicles were bound to latex beads (Fig S4B). The latter explained the inhibitory effects of the anti-GM-CSF antibody. The inhibitory effect was less pronounced with GM-CSF–stimulated cells, likely because the antibody was not effectively neutralizing the recombinant GM-CSF.

Recombinant *E. coli*–derived GM-CSF was used to generate the monocyte-derived DC, from which the EV were originally derived. Hence, this exogenously added cytokine could have been carried along in the course of the EV purification. Since bacterial-derived GM-CSF, unlike eukaryotic GM-CSF, is not glycosylated, both forms can be distinguished by Western blot. Both imDC-EV and maDC-EV contained substantial amounts of glycosylated GM-CSF (26 kD), which was, as expected, also detected in cell lysates of the producer maDC (Fig 3C, red box). Conversely, only the cell lysates from the producer DC, but not DC-EV, revealed traces of the recombinant GM-CSF (blue box). This result formally excluded a carryover of exogenously added GM-CSF, and hence DC-derived EV contained only endogenously produced GM-CSF.

While imDC-EV and maDC-EV had similar GM-CSF content as measured by Western blot (Fig 3C), only maDC-EV induced a DC-typical phenotype in monocytes. We therefore examined maDC-EV for additional CCF by a protein array and multiplex technology. We found an abundance of additional factors, and many of

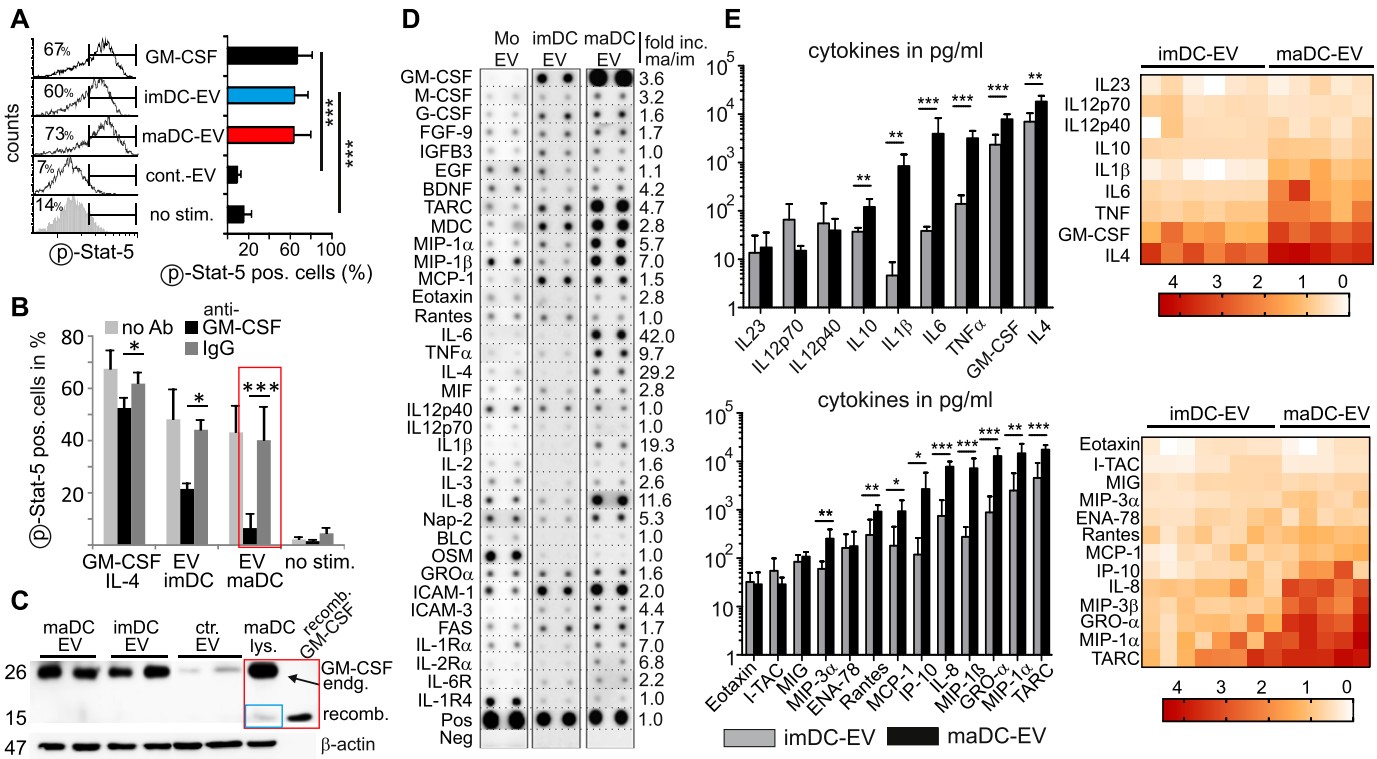

**Figure 3. DC-EV induce GM-CSF signaling and convey a cornucopia of effector molecules.**
**(A)** DC-EV induce Stat5 phosphorylation. Peripheral monocytes (2 × 10⁵) were incubated with EV (10 μg) derived from imDC and maDC and 293T cells (control EV) or stimulated with GM-CSF/IL-4 (each for 15 min) or left untreated. Subsequently, cells were fixed and analyzed for Stat5 phosphorylation by intracellular FACS. FACS blots depict one representative experiment. Three healthy donors were analyzed to calculate the mean and SEM. **(B)** Anti-GM-CSF blocks DC-EV–induced Stat5 phosphorylation. Same experimental setup as in (A); however, one cell aliquot of each culture was left untreated or was supplemented with anti-GM-CSF. Triplicate cultures were performed for each donor (three donors) to calculate the mean and SEM. **(C)** DC-EV–derived GM-CSF is derived from the producer DC. Lysates of purified DC-EV and control EV (from 293T cells) was blotted for endogenous (endg.) GM-CSF using lysates of maDC (maDC lys.) and recombinant (recomb.) GM-CSF as control. **(D, E)** DC-derived EV contain multiple CCF. **(D)** EV were collected from monocytes and monocyte-derived imDC and maDC (50 μg) and subsequently analyzed for the indicated factors using commercially available protein arrays (RayBiotech). The pixel intensity of each dot was determined by ImageJ, and the value was adjusted in relation to the internal positive control, which was set to 1. Shown is one representative analysis performed with four different donors (see also Fig S4C). **(E)** Same experimental setup as in (D); however, the EV contents were analyzed using bead-based quantitative immunoassays (BioLegend). imDC-EV were analyzed from six and eight different donors for cytokine and chemokine content, respectively (gray columns). maDC-EV from six different donors were analyzed (black columns). Heat maps depict the common logarithm (log(10)) of the cytokine and chemokine concentrations of each individual sample. Bar graphs indicate mean values ± SEM. Statistical significance was analyzed by the t test: *P < 0.05, **P < 0.01, and ***P < 0.005.
Source data are available for this figure.

them strongly up-regulated in maDC-EV compared with imDC-EV or monocytes. This was demonstrated by a fold increase in a protein array (Fig 3D, second example in Fig S4C) or by protein concentration (pg/ml) through multiplex technology (Fig 3E), also displayed and summarized by heat maps. Up-regulated factors included IL-1β, IL-6, TNF, IL-4, GM-CSF, IL-8, MIP-3β, GROα, MIP-1α, and TARC. To confirm that these factors were vesicle associated, maDC-derived vesicles were purified by an iodixanol (OptiPrep) gradient and each density fraction was analyzed for selected CCF by multiplex technology. Indeed, fractions that contained EV, as confirmed by electron micrographs, also harbored the CCF (Fig S5). This demonstrated that the assortment of factors measured in Fig 3D and E were indeed associated with vesicles. Taken together, the rich CCF content of maDC-EV was likely involved in the differentiation process of monocytes towards imDC and hinted at additional functions and/or target cell effects of these vesicles.

## Murine BM–derived DC-EV behave similarly as their human counterparts

We sought ways to analyze DC-derived EV in vivo. We wondered whether EV would attract monocytes (CD11b⁺/Ly6C⁺/Ly6G⁻) and/or other immune cells, as implied by the presence of chemokines in these vesicles. In addition, we asked whether DC-EV would induce differentiation processes, as implied by our in vitro experiments. To this end, we first confirmed that EV derived from murine BM–derived DC (BMDC-EV) behaved similarly as EV from human DC. Because the human standard MC is not suitable to mature murine DC, we used Poly I:C and LPS instead. Both, EV from immature and mature BM-derived DC (imBMDC-EV and maBMDC-EV), were efficiently taken up by murine monocytes and only to a lesser degree by B cells and granulocytes (Fig S6A). Like in human maDC-derived EV, an array of cytokines and chemokines were found in mature

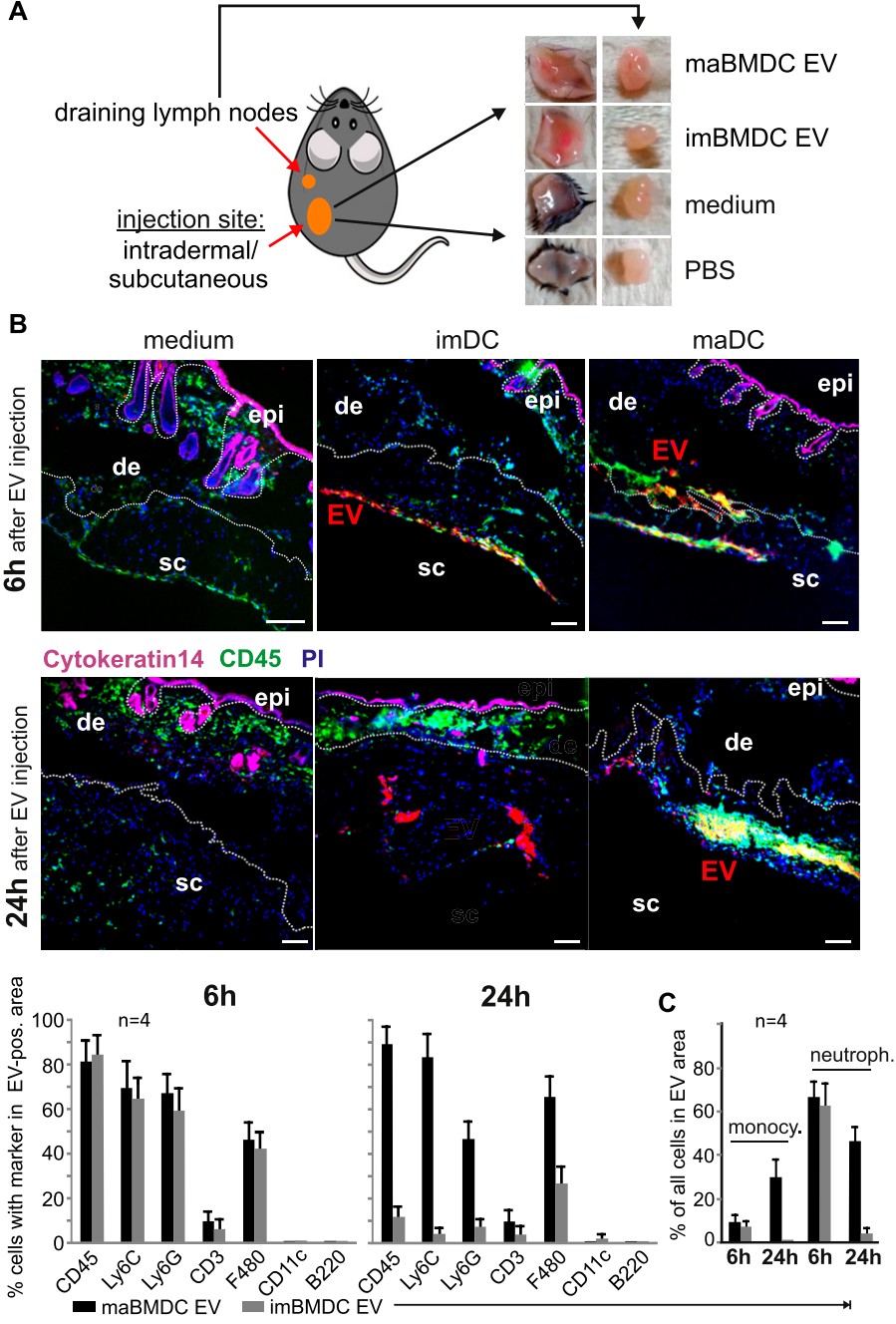

**Figure 4.   Skin-injected BMDC-EV attract immune cells.**
**(A)** Cartoon depicting the injection site of PKH26-labeled BMDC-EV, and images from excised skin patches and draining lymph nodes used for MELC analysis and marker quantification. For control, PKH26-containing medium and PBS were injected. **(B)** BMDC-derived EV attract immune cells in the skin. Tissue sections from skin patches described in (A) were subjected to a MELC analysis. Images represent an overlay of four markers (CD45, cytokeratin-14, PI, and EV). Cytokeratin-14 and CD45 were stained by antibodies, whereas EV (red stain) were visualized through PKH26. Tissue sections from two animals (6 h and 24 h) are presented. Using the StrataQuest software, the relative presence (in percentage) of common immune cell markers was quantified in EV-containing tissue areas. The relative presence of cells (percentage of cells with marker in EV areas) is depicted by a bar diagram as indicated. Note: individual images for these markers are presented in Fig 5. Quantifications of MELC analyses from four different injection sites were used to determine the SEM. Scale bars represent 100 μm. **(C)** Monocytes and neutrophils in imDC-EV and maDC-EV areas. Monocytes (CD11b$^+$/Ly6C$^+$/Ly6G$^-$) and neutrophils (CD45$^+$/Ly6C$^+$/Ly6G$^+$) were quantified in the EV areas using the StrataQuest software as explained in Table 1 and Fig S8A. epi, epidermis; de, dermis; sc, subcutaneous.

(ma) BMDC-EV, particularly after stimulation with Poly I:C (Fig S6B and C). A noticeable difference was the presence of IL-12p70, which was absent in the human maDC-EV. This was likely due to the presence of PGE$_2$ in the human MC, inhibiting the production of IL-12p70 (Kalinski et al, 1997), while the Poly I:C used for the murine BMDC facilitates its production.

Like with human cells, maBMDC-EV induced a morphology in murine monocytes that resembled that of GM-CSF–induced BMDC and resulted in the up-regulation of CD11c and MHC-II expression (Fig S7). Hence, maBMDC-EV could be used to analyze their function in the mouse model.

## Skin-injected BMDC-EV attract immune cells

To mimic the secretion of EV from DC in vivo, PKH-labeled imBMDC-EV and maBMDC-EV, as characterized in Fig S6, were injected into the skin of mice. The skin patches around the injection sides, as well as the draining lymph nodes, were obtained 6 and 24 h after injection, showing a red color (Fig 4A). Sections of these tissues were analyzed by immunofluorescence using the multiepitope ligand cartography (MELC) technology, which allows sequential staining of the same tissue section by multiple antibodies, as demonstrated recently (Ostalecki et al, 2017). In addition, we employed improved

**A**

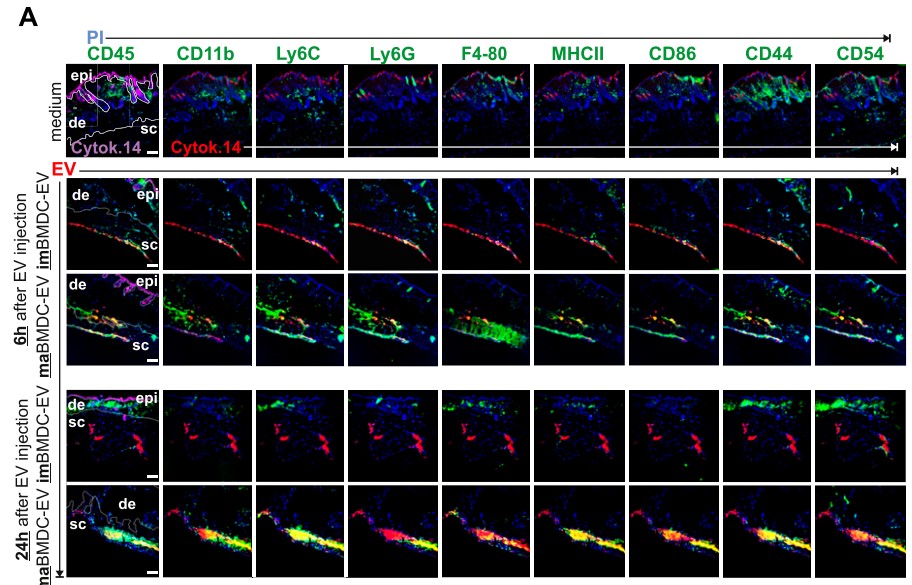

**Figure 5. Immune cells attracted by maBMDC-EV express myeloid activation markers (see also Fig S7).**
**(A)** Individual images of a MELC analysis, assessing immune cell markers in BMDC-EV–containing skin tissue areas. The same skin tissue sections shown in Fig 4B from imBMDC-EV–, maBMD-EV–, and medium-injected areas, obtained after 6 h and 24 h, were analyzed for the indicated markers (green) and EV colocalization (yellow; see also Fig S7). Note: for better orientation, the images on the left were duplicated from Fig 4B. **(B)** Co-expression of myeloid activation and differentiation markers with monocytes (CD11b+/Ly6C+/Ly6G−). Using the StrataQuest software, monocytes were identified (see Fig 4C) and analyzed for co-expression of the indicated markers in the EV-containing tissue area, expressed in percentage of total monocytes found in the EV area. The analysis was performed for each time point in tissue sections from four different injection sites. The obtained numbers were used to determine the SEM. Scale bars represent 100 μm.

**B**

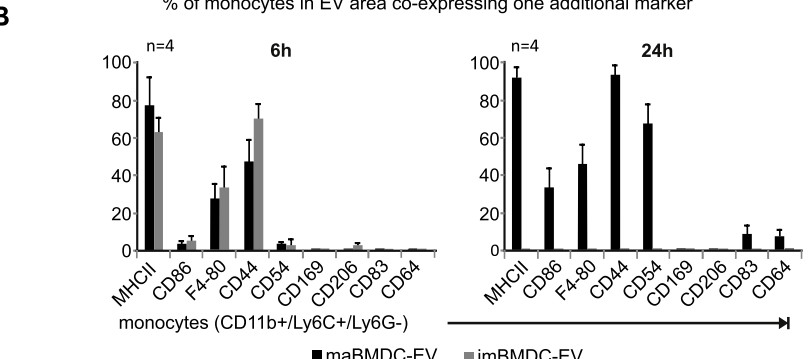

% of monocytes in EV area co-expressing one additional marker

imaging software (StrataQuest from TissueGnostics), able to quantify multiple immunofluorescence markers in tissue (Fig S8A). By combining both analysis systems, we were able to demonstrate as well as quantify multiple markers in tissue areas harboring EV. EV-containing tissue areas were demarcated by the software as explained in Fig S8A. To confirm that the injected EV were ingested by cells and not merely deposited, confocal images were taken from tissue sections showing EV granules in the cytoplasm and perinuclear region of CD11b+ cells (Fig S8B). Aside from control injections (PKH26 and PBS), four of these areas (examples in Fig S9) were analyzed for each time point (6 h/24 h) and each EV type (EV from im-/maDC) in four different animals. In Figs 4, 5, 6, and 7 and Table 1, one representative example for each time point is shown and analyzed.

After 6 h, both imBMDC-EV– and maBMDC-EV–containing skin areas roughly contained the same number of cells/$\mu m^2$ (~1 cell in 18 $\mu m^2$, Table 1). In both EV areas, many cells expressed the immune cell markers CD45, Ly6C, Ly6G, and F4-80, to some extent CD3, but not CD11c or B220 (Figs 4B, 6 h images and graph; note: except for CD45, all images for individual markers are shown in Figs 5A and S10). Many of these cells were neutrophils (CD45+/Ly6C+/Ly6G+: ~60–75%). Classical monocytes (CD11b+/Ly6C+/Ly6G−: ~7–11%) were also present but not

prominent (Fig 4C and Table 1, orange box). No immune cells were found in locations injected with PKH26-containing medium (Figs 4A, B, and 5A, and S10) or PBS (data not shown).

After 24 h, the cell concentration/$\mu m^2$ in both areas remained similar (~1 cell in 18 $\mu m^2$, Table 1); however, only maBMDC-EV areas contained high proportions of cells positive for CD45, Ly6C, Ly6G, and F4-80, whereas almost no immune cells were found in the imBMDC-EV areas (Figs 4B, 24 h images and graph, Figs 5A and S10). Notably, in maBMDC-EV areas, monocytes were now more abundant (~25–35%), while the proportion of neutrophils decreased but remained high (~42–48%) (Fig 4C and Table 1, yellow box). To verify the presence of monocytes in these densely packed tissue areas, they were also demonstrated by confocal imaging of tissue sections (Fig S11A). Taken together, both types of EV at first (6 h) attracted myeloid cells, particularly neutrophils, to a similar extent. However, only when maBMDC-EV were present, these cells remained at the EV deposition site (at 24 h) and/or migrated into this area, including a significant number of monocytes.

An analysis of myeloid activation and differentiation markers, which had been stained in the same MELC analysis run, revealed that the monocytes attracted to EV areas at 6 h co-expressed only few activation and differentiation markers, including MHC-II, F4-80,

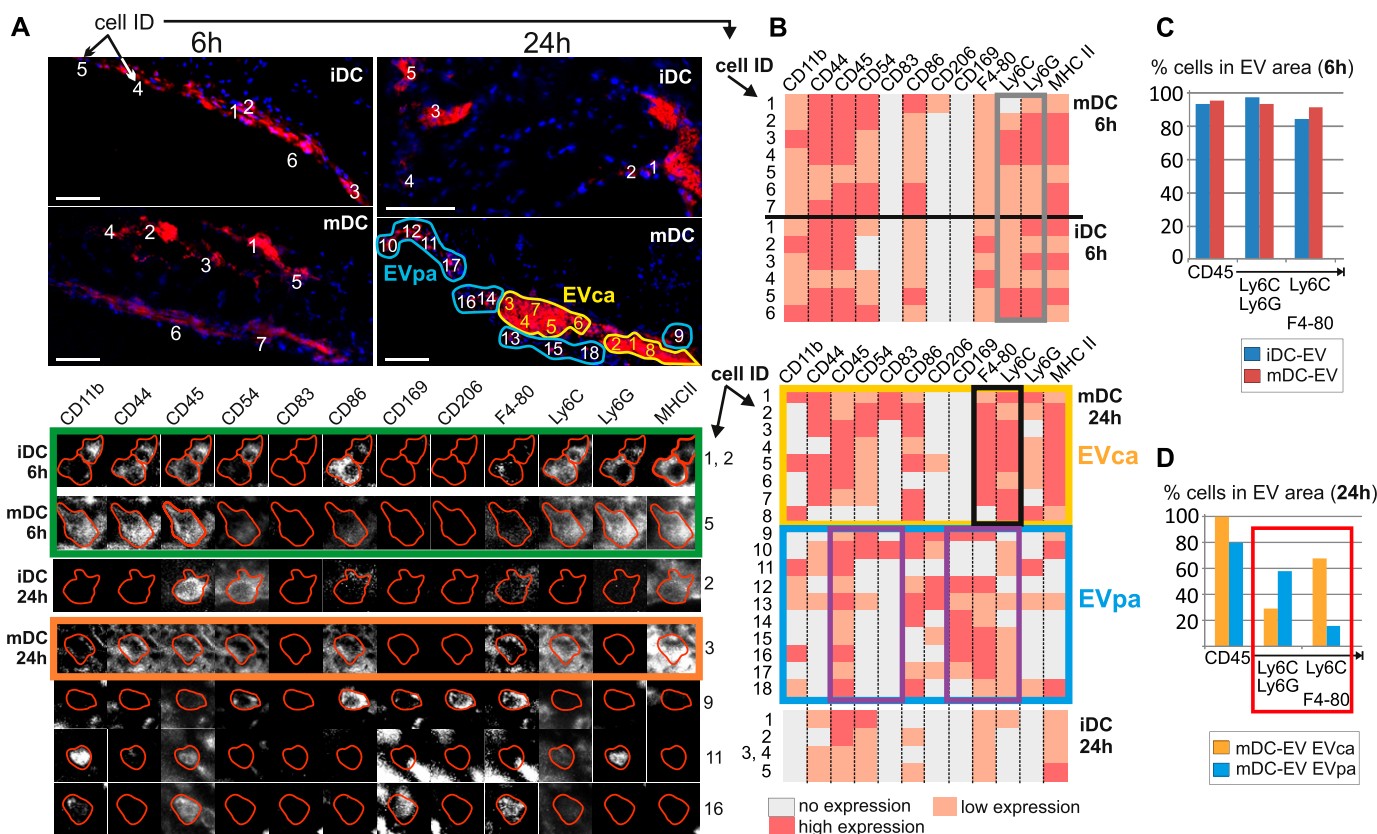

**Figure 6. BMDC-EV induce phenotypically different myeloid cell sub-populations.**
**(A)** Protein expression profiles of individual immune cells in imBMDC-EV- and maBMDC-EV–containing tissue areas 6 h and 24 h after injection. The upper panels (also analyzed in Figs 4 and 5) show the whole tissue areas, whereas the lower panels depict the MELC protein profile of individual cells from these areas. Individual cells were chosen randomly and assigned an ID number (cell ID). Cells were selected in areas where EV were concentrated (EVca; yellow demarcated area, upper panels) or less concentrated (EVpa; blue demarcated areas). The latter was determined by the immunoreactivity score (IRS) (Remmele & Stegner, 1987). Colored boxes were inserted for explanations in the main text. Scale bar represents 50 μm (upper panels) and 7.5 μm (lower panels). **(B)** Expression levels of indicated markers for each numbered cell (cell ID in [A]). For better understanding, protein expression was divided into three levels (no, low, and high expression), which were color-coded. **(C, D)** Relative presence of cells with triple marker combinations in imBMDC-EV- and maBMDC-EV–containing tissue areas shown in (A). Triple combinations of the most abundant markers (CD45, Ly6C, Ly6G, and F4-80) and CD169 (serving as internal control) were assessed by StrataQuest software as explained in Fig S8 and Table 1 and displayed in bar diagrams.

and CD44 (Fig 5B). Little or no co-expression was recorded for CD86, CD54, CD169, CD206, CD83, CD64 (Fig 5B), and additional markers shown in Fig S10. After 24 h, monocytes harbored additional activation/differentiation markers including CD86, CD54, CD83, and CD64 (Figs 5B and S10). Furthermore, cells positive for CD3, CD206, and CD169 (Fig S10, white boxes) were present. Conversely, few immune cells were found in association with imBMDC-EV. Together, this suggested the potential initiation of differentiation processes only in areas with maBMDC-EV. In line with this assumption, we noticed that in maBMDC-EV depositions (24 h) the identified markers were not evenly distributed but clustered in different areas (Fig 5A, lower row of image panels), implicating the presence of differently evolving or developing cell populations.

## BMDC-EV induce phenotypically different myeloid cell sub-populations

To substantiate this assumption, we determined the topographical marker composition in the imBMDC-EV– and maBMDC-EV–injected tissue areas at 6 and 24 h (Fig 6A, upper panels). This was done by assessing the protein expression profiles of representative cells (see numbered cells [cell ID] in Fig 6A), exemplified in the lower image panels of Fig 6A. For better overview, expression levels were divided into no, low, or high expression by color coding and summarized for all demarcated cells in Fig 6B. In compact tissue, a clear assignment of multiple markers to individual cells is difficult when their expression levels are high, potentially leading to false-positive signals. To avoid misinterpretations, we concentrated on general shifts in marker expression.

After 6 h, the surface marker profile of infiltrating cells was comparable in both EV locations and typically positive for CD44, CD45, CD54, Ly6C, Ly6G, and MHC-II and negative for CD83, CD11c, CD169, and CD209. Examples are given in Fig 6A (green box) and summarized in Fig 6B (top panel). Supporting this observation, cells with triple combinations of the most abundant markers (CD45, Ly6C, Ly6G, and F4-80), analyzed by software for all cells in the EV area, were equally present (Fig 6C). Hence, after 6 h, there was no indication of marker variation in the infiltrating immune cell population.

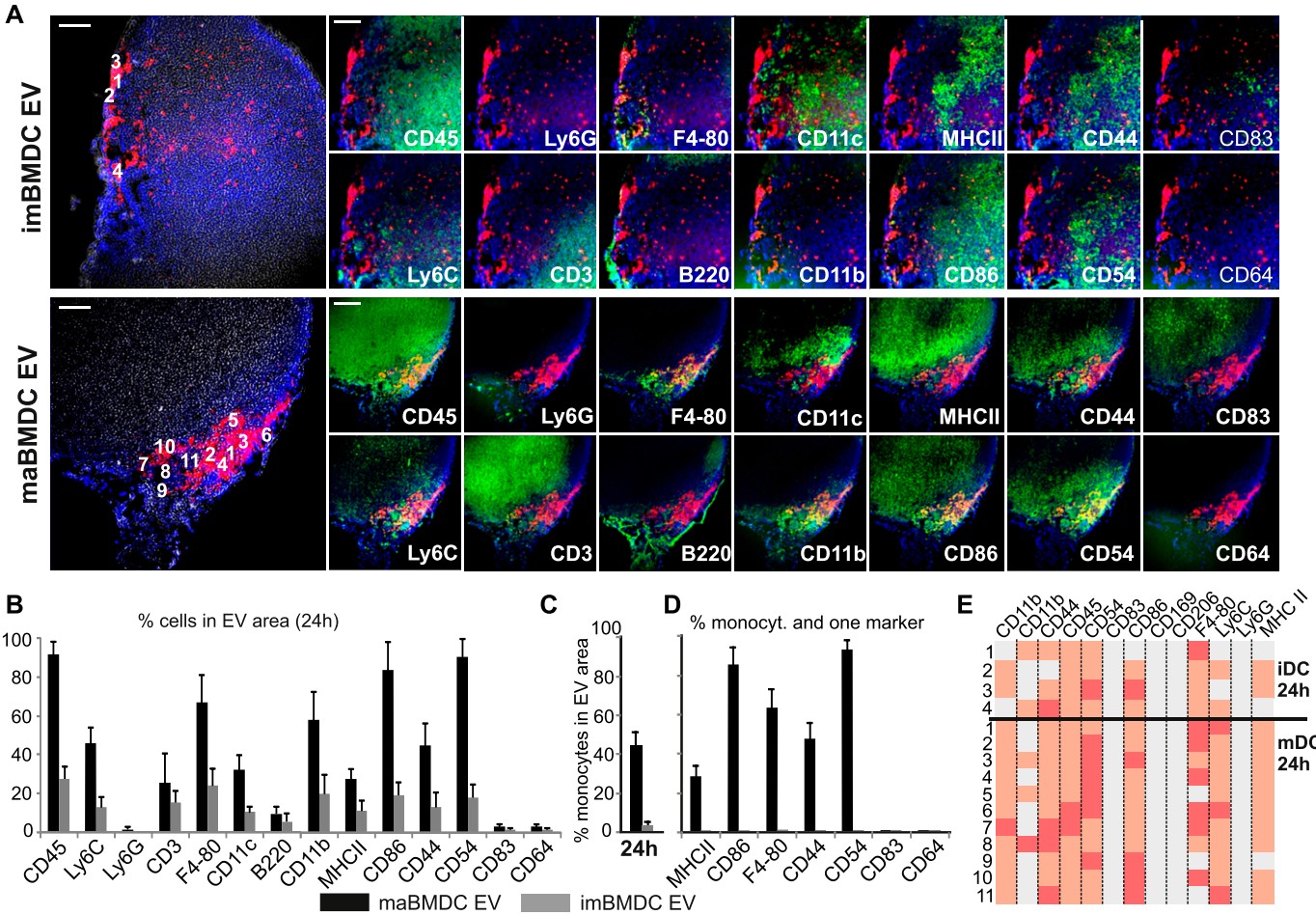

**Figure 7. BMDC-EV attract immune cells in lymph nodes similarly as in the skin.**
**(A)** Individual images of a MELC analysis, assessing EV and immune cell markers in lymph nodes. Tissue sections from draining lymph nodes after skin injections of imBMD-EV and maBMD-EV (red label) obtained after 24 h were analyzed by MELC for the indicated markers and for colocalization with EV. Numbers in the left-most panels depict the localization of cells analyzed in (D). **(B)** The relative presence of cells in EV-containing areas (in percentage) was assessed by StrataQuest software as in Figs 4 and 5. The analysis was performed in tissue sections from four different lymph nodes. The respective numbers served to calculate the SEM. **(C)** Monocytes in imDC-EV and maDC-EV areas. Monocytes (CD11b$^+$/Ly6C$^+$/Ly6G$^-$) were quantified in the EV areas using the StrataQuest software as explained in Table 1 and Fig S8A. **(D)** Identified monocytes were analyzed for co-expression of additional markers, expressed in percentage of total monocytes found in the EV area. The analysis was performed in tissue sections from four different injection sites. The obtained numbers were used to determine the SEM. Scale bars represent 100 μm. **(E)** Expression levels of indicated markers for each cell numbered in (A). For better understanding, protein expression was divided into three levels (no, low, and high expression), which were color-coded as in Fig 6B. Scale bars represent 100 μm.

After 24 h, cells expressing these markers seemingly persisted, but only in areas with high maBMDC-EV concentrations (yellow demarcated EV-concentrated area [EVca] in Fig 6A and B). The immunoreactivity score for the EV stain in the EVca was 9–12 (strong) (Remmele & Stegner, 1987). However, now most of the cells displayed a Ly6C/F4-80–positive phenotype rather than a Ly6C/Ly6G-positive phenotype (likely neutrophils) (Fig 6B, compare gray and black boxes). The Ly6C/F4-80 phenotype was confirmed by confocal imaging of a tissue section from this area (Fig S11B).

Cells located in EV peripheral areas (EVpa in Fig 6A and B, blue box) with a lower density of maBMDC-EV (immunoreactivity score 6–8: moderate) displayed a more heterogeneous cell phenotype, which was in general positive for F4-80/CD169 and/or CD206 and negative for CD44/CD54 (Fig 6B, magenta boxes). Hence, in EVpa areas, cells were more likely to be double positive for Ly6C/Ly6G and less likely to

be positive for Ly6C/F4-80. This was confirmed by the software-based assessment of all cells in the maBMDC-EV areas (Fig 6D, red box). In addition, we recorded more monocytes in EVca areas (36.8%) as compared with EVpa areas (10.8%) (Table 1). In imBMDC-EV areas, all these cell phenotypes were absent (Fig 6B, lower panel). Together, these results suggested that the tissue concentrations of maBMDC-EV correlated with different or evolving myeloid cell sub-phenotypes.

**maBMDC-EV attract immune cells in lymph nodes similarly as in the skin**

A substantial amount of skin-injected BMDC-EV reached the draining lymph node, and particularly maBMDC-EV induced a slight swelling as judged by naked eye (Fig 4A). Analysis of tissue sections confirmed that both imBMDC-EV and maBMDC-EV

**Table 1.  Cells with monocyte surface markers in EV-containing tissue areas (skin).**

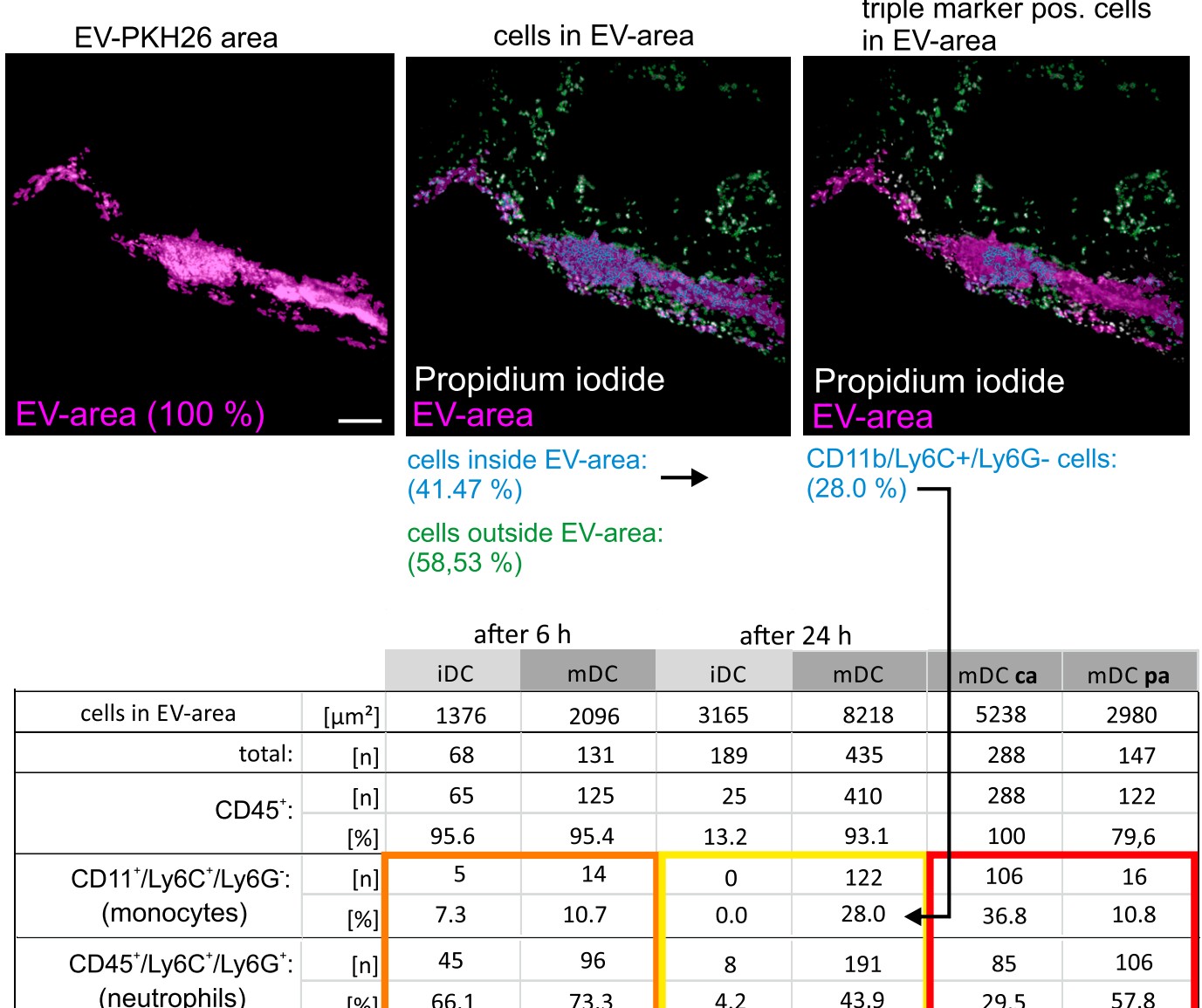

| | | after 6 h | | after 24 h | | | |
|---|---|---|---|---|---|---|---|
| | | iDC | mDC | iDC | mDC | mDC **ca** | mDC **pa** |
| cells in EV-area | [μm²] | 1376 | 2096 | 3165 | 8218 | 5238 | 2980 |
| total: | [n] | 68 | 131 | 189 | 435 | 288 | 147 |
| CD45⁺: | [n] | 65 | 125 | 25 | 410 | 288 | 122 |
| | [%] | 95.6 | 95.4 | 13.2 | 93.1 | 100 | 79,6 |
| CD11⁺/Ly6C⁺/Ly6G⁻: (monocytes) | [n] | 5 | 14 | 0 | 122 | 106 | 16 |
| | [%] | 7.3 | 10.7 | 0.0 | 28.0 | 36.8 | 10.8 |
| CD45⁺/Ly6C⁺/Ly6G⁺: (neutrophils) | [n] | 45 | 96 | 8 | 191 | 85 | 106 |
| | [%] | 66.1 | 73.3 | 4.2 | 43.9 | 29.5 | 57.8 |

Using StrataQuest software, cells were identified and quantified in EV-containing areas through propidium iodide (PI) assessment (41.47% of all cells were found in EV area, first two images). Subsequently, MELC images for monocyte markers (CD11b⁺/Ly6C⁺/Ly6G⁻) were superimposed and colocalizing signals were assessed (EV and markers: 28%). A similar approach was taken to assess neutrophils (CD45⁺/Ly6C⁺/Ly6G⁺) and monocytes in EVca and EVpa areas described in Fig 6A. Scale bars represent 100 μm.

reached lymph nodes to a similar extent (Fig 7A) and cells associated with these areas in a comparable concentration (~1 cell in 18–22 μm², Table S1).

After 24 h, the maBMDC-EV areas were dominated by cells expressing CD45, Ly6C, and F4-80. In addition, CD11c- and CD3-positive cells were present, but no neutrophils (Ly6G⁺) and few B cells (B220⁺). Notably, most cells (80–90%) stained for the myeloid activation markers CD86 and CD54 (Fig 7A). Within this cell population, around 42% expressed the marker combination for monocytes (Fig 7C and Table S1), and many if not most of the monocytes co-expressed activation and differentiation markers

(CD86, F4-80, CD44, and CD54) similar as seen in the skin (Fig 7D, compared with Fig 5B).

In imBMDC areas, significantly fewer immune cells were detected and almost no cells with monocyte markers (Fig 7A–D and Table S1). Otherwise, the individual marker combination of cells was comparable in both EV areas as judged by the analysis of individual cells (Fig 7A, left panels, and Fig 7E). Taken together, maBMDC-EV attracted immune cells in a similar fashion as in the skin, albeit there seemed to be a predominance of monocytes with activation and differentiation markers and a higher proportion of CD11c⁺ and also CD3⁺ T cells.

# Discussion

Here we demonstrate that EV shed by maDC stimulate the differentiation of monocytes towards imDC in vitro and potentially in vivo, as suggested by the development of DC-typical functional properties and surface marker expression. This differentiation process depended on effector molecules present in these vesicles, including GM-CSF, which is commonly required for the differentiation of monocytes into DC in vitro (Inaba et al, 1992; Lutz et al, 2017). However, the cornucopia of effectors found in maDC-EV pointed to more complex functions and target cell effects, and blocking experiments with anti-GM-CSF may have inhibited the internalization process of EV all together. At least one of these functions was revealed when murine BMDC-EV were injected into mouse skin. The vesicles attracted predominantly myeloid cells and to some extent T cells, potentially through chemokines such as MCP-1 and TARC. In addition, they likely initiated activation and differentiation processes in monocytes. The latter remained an assumption, as the development of imDC from monocytes likely takes up to 6 d, and hence longer than the presence of labeled EV could be recorded (data not shown). However, several findings support this conclusion. This includes (1) the appearance of monocytes bearing additional activation and differentiation markers, including MHC-II, CD86, F4-80, CD44, CD54, CD169, CD206, CD83, and CD64, (2) the absence of these cells after injection of imBMDC-EV or medium/PBS, (3) the beginning flux of CD3 T cells into EV areas in the lymph node and skin (Figs 5 and 7), and (4) previous reports demonstrating the development of monocytes into DC and macrophages in mouse models (Naik et al, 2006; Cheong et al, 2010; Zigmond et al, 2012; Menezes et al, 2016).

Monocytes treated with maDC-EV expressed a number of markers that are not found on classical monocyte-derived imDC, including CD64, Zbtb46, and CD14. In addition, the cells were positive for 6-sulfo LacNAc (slan) and negative for CD1c, both characteristics of circulating slan$^+$ imDC (Schakel et al, 2002). However, the here-described cells are likely not identical with slan$^+$ imDC, as they maintained CD14 expression, a typical monocyte marker, which is also found on inflammatory DC (Segura et al, 2013). CD14 has regulatory function in infection and damage (Zanoni et al, 2017), a necessary requirement for immune cells in inflamed tissue. Potentially, the array of factors found in DC-EV induced a more complex differentiation process than described here. In summary, we suggest that maDC-EV–differentiated monocytes, or EV-induced imDC, belong to a growing spectrum of monocyte-derived inflammatory cells, and perhaps more specifically to a spectrum of slan$^+$ imDC.

In line with this conclusion, the skin-infiltrating immune cells 6 h after EV injection displayed a marker profile that seemed to some degree similar to what has been described for mouse (inf) DC (MHC-II$^+$, CD11b$^+$, CD11c$^+$, F4-80$^+$, CD206$^+$, CD64$^+$, and Ly6C$^+$) (Segura et al, 2013). However, several (inf)DC-defining markers were low or negative including CD11c, CD206, and CD64 (Fig S10). The lack of these markers is likely due to the fact that (1) the infiltrated cells described here represent immature precursors and (2) the assumed differentiation process was not completed after 24 h.

Recruitment of immune cells into tissue and the draining lymph node by DC-derived EV may serve to increase and perpetuate an immune reaction, at least as long as there is a maturation stimulus present. Such a scenario, with an ongoing generation of inflammatory DC from monocytes, has been originally described for Leishmania infection (Leon et al, 2007). Based on these and related findings, a "wind mill" model was proposed, describing the perpetuation of an immune response with the help of monocyte-induced DC (Lutz et al, 2017). Like most models in immunology, these mechanisms are based on the extracellular secretion of cytokines, chemokines, and other soluble factors by immune cells. However, secreted factors may rapidly dilute in extracellular space and fail to reach critical concentrations and/or proper conformations (e.g., TNF trimer) to attract and/or differentiate new immune cells in sufficient numbers. EV with their rich factor content, and a presumed monocyte-targeting mechanism, are likely a more efficient mediator of these functions, with less off-target effects. This directed target cell effect has been demonstrated in vitro and in tissue by us and others with TNF vesicles and vesicular structures also termed "focal TNF" (Ostalecki et al, 2016; Yuan et al, 2017). In line with this assumption, we found that about 170-fold less of EV-associated GM-CSF (2.5 ng) is required than free GM-CSF to differentiate monocytes into DC in vitro (Fig S2A).

The rather high amount of EV injected in our experiments may not reflect the individual steps of this model, in particular an assumed self-perpetuating increase of the immune reaction starting from few maturing DC. However, we were able to visualize the principles of this mechanism. The fact that only EV from maDC induced a lasting attraction and differentiation of immune cells was of particular importance and allows us to propose a variation of the "wind mill" model as depicted in Fig 8A. It is easy to imagine that such a self-perpetuating mechanism could lead to chronic inflammatory conditions as reported for slan$^+$ DC (Hansel et al, 2011, 2013).

We demonstrated that the EV-derived effectors induced signaling events in target cells (Fig 3A). How and where these events are executed is not clear yet. Analyzing the secretion and signaling mechanism of EV-contained TNF, we have previously suggested a sub-membrane/intracellular signaling mechanism that is initiated by the fusion of incoming vesicles, containing mature TNF ligands in their lumen, with endosomal structures that contain their respective receptors (Fig 8B). In the here-adapted model, it is assumed that GM-CSF is matured/processed and packaged in a similar manner. Hence, we would assume that at least a sizable fraction of GM-CSF is transferred within the vesicular lumen of EV to monocytes, potentially through receptor-dependent endocytosis (Fig 8B). However, alternative explanations are imaginable, as, for example, the concentrated presence of GM-CSF or other ligands on the outer membrane of EV, as suggested recently (Tkach et al, 2017), that may origin from multivesicular bodies and stimulate GM-CSF receptors on the outer membrane (Fig 8B). In this case, the GM-CSF receptor may also function as a docking site for these vesicles. In line with this assumption, we found at least some portion of GM-CSF on the outer membrane of the EV (Fig 3B).

As implicated by our model, each single EV may contain a different effector, each starting its respective signaling cascade in or on the target cell. Hence, the differentiation of the target monocyte

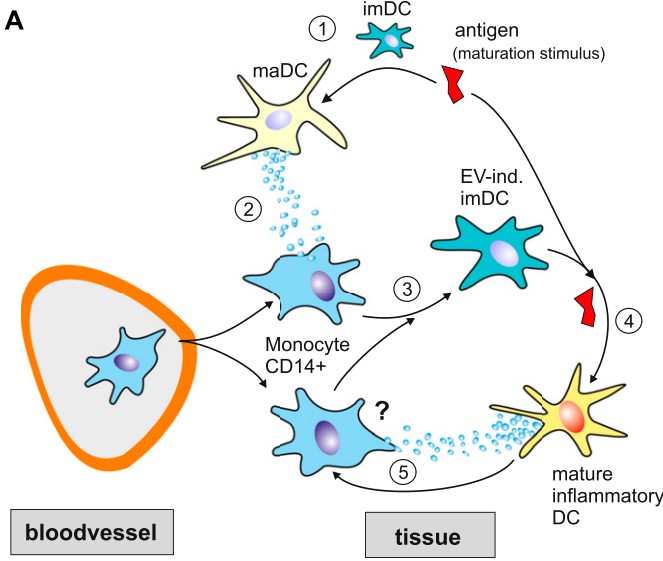

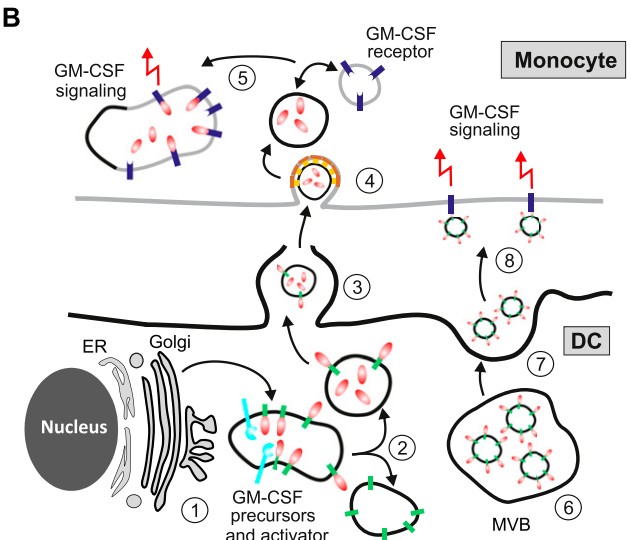

**Figure 8. Speculative models of monocyte-derived imDC generation through EV-induced signaling in endosomal compartments.**
**(A)** Generation of monocyte-derived DC through DC-derived EV ("wind mill" model). (1) An imDC receives a maturation stimulus and differentiates into a maDC. (2) The resulting maDC secretes CCF-containing EV, which (3) attract and differentiate monocytes into imDC. (4) In case the maturation stimulus (danger signal) is still present, this DC precursor develops into a mature inflammatory DC. (5) It is assumed (not shown in this manuscript) that the mature inflammatory DC produces more EV and hence perpetuates this cycle. **(B)** GM-CSF signaling through EV. We depict two principle mechanisms of EV-mediated GM-CSF signaling, a mechanism that suggests sub-membrane signaling (steps 1–5) and outer membrane–mediated signaling (6–8). EV-mediated sub-membrane signaling: (1) GM-CSF precursor proteins are produced in the ER/Golgi and packaged into endosomal compartments, and, upon an activation stimulus (not shown), fuse with compartments containing effector proteases (activator). (2) This leads to the maturation and sequestration of GM-CSF into EV and subsequently (3) secretion of these vesicles through the membrane as shown in Muratori et al (2009). Both steps have been analyzed for the TNF precursor protein (Ostalecki et al, 2016). In the model here, we assume that GM-CSF is also incorporated into the membrane of EV as demonstrated by FACS in Fig S4B. (4) These vesicles are preferentially ingested by monocytes and (5) fuse with endosomal compartments that contain GM-CSF receptors. This leads to GM-CSF–signaling events from endosomal compartments

would depend on the number and array of signaling cascades initiated by incoming EV. Since the EV content and concentration may vary topographically, different cell sub-populations may arise as demonstrated in Figs 6 and 7. Whether these described sub-populations represent snap shots of cells in the course of a differentiation process or represent sub-populations with different functions is not clear.

Using quantitative proteomic analysis, others have not found the rich CCF content we report here (Kowal et al, 2016). Despite being a powerful technique, application of mass spectrometry (MS) to CCF analysis is challenging. First, MS is able to detect only charged (ionized) peptides, but not all peptides are sufficiently ionized. Second, proteins with low molecular weight (like CCF) and low abundance are likely not detected without enrichment or pre-fractionation. Third, the sensitivity of shotgun MS is usually insufficient for detection of low abundant proteins in complex biological samples (Kupcova et al, 2017). Hence, we would assume that quantitative proteomics is not a suitable technology to analyze low abundant CCF in EV preparations.

The here-described effects of DC-EV on immune cells may provide an additional perspective on many established immune mechanisms, and the effect of DC-derived EV may not be restricted to monocytes alone. More insight into the content of EV and their target cell effects, including sub-membrane signaling, is required before their relevance in immune regulation is fully understood.

# Materials and Methods

### Protein assays

#### Western blot
Proteins separated by SDS-PAGE were transferred onto nitrocellulose filters (Schleicher & Schuell) using the wet blotting device "Mini-Protean II Cell and System" (Bio-Rad) at 400 mA for 45 min. Filters were immersed in blocking buffer for 1 h at RT. After three washes with distilled water, primary antibody diluted 1:500–1:5,000 in TBST was added and incubated for 1 h (RT) or overnight (4°C). Thereafter, filters were washed 3 times for 5 min with PBS/0.02% Tween20 before being incubated for 1 h at 4°C with a secondary HRP–conjugated anti-mouse or anti-rabbit antibody diluted 1:2,000–1:5,000 in PBS/0.02% Tween20/5%. Finally, the filters were washed three times for 10 min with PBS/0.02% Tween20, and protein bands were visualized by ECL (Pierce) according to the manufacturer's instructions.

#### Human CCF array
Purified EV (through differential centrifugation, see below) were applied to the RayBio Human Cytokine Array C-S (Hölzel Diagnostika, AAH-CYT-1000-2) according to the manufacturer's instructions. Cytokines were identified based on Table S1. Positive results were quantified by ImageJ, using the internal control standards as reference.

(sub-membrane). EV-mediated outer membrane signaling: (6) EV that have GM-CSF predominantly on the EV membrane are accumulating in multivesicular bodies (MVBs). (7) These MVBs fuse with the outer membrane of the cell and release the EV. (8) In extracellular space, these EV attach to the next GM-CSF receptor, e.g., of a neighboring cells, and induce GM-CSF signaling from the outer membrane.

## Measurement of EV cytokine content

Cytokines and chemokines were measured and quantified in EV preparations (10 $\mu$l) using bead-based immunoassays according to the manufacturer's instructions, all from BioLegend (Human T Helper Cytokine Panel, 740001; Human Cytokine Panel 2, 740102; Human Proinflammatory Chemokine Panel, 740003; Mouse Inflammation Panel, 740446).

## Heat maps

Heat maps were generated using GraphPad Prism, Version 7.00 (GraphPad Software), from common logarithms of the concentrations in pg/ml for each individual sample.

## Cells

### Generation of PBMCs

PBMCs from healthy volunteers were obtained following approval by the local ethics committee and informed consent. Leukoreduction system chambers were obtained after plateletpheresis. The resulting platelet-free cell sample was diluted 1:2 in PBS, and the PBMC-containing buffy coat was isolated after density gradient centrifugation on Lymphoprep (Axis-Shield 1114544) at 500 $g$ for 30 min at room temperature. PBMCs were then washed three times in PBS/1-mM EDTA: first wash: 282 $g$, 15 min, 4°C; second wash: 190 $g$, 10 min, 4°C; third wash: 115 $g$, 12 min, 4°C.

### Generation of DC

Monocyte-derived DC were generated from PBMCs as described previously (Thurner et al, 1999), using GM-CSF and IL-4 (6 d) to generate imDC and a MC (IL-1$\beta$, IL-6, TNF-$\alpha$, and PGE$_2$) to obtain maDC on day 7 (IL-4 was also from Strathmann, Hamburg, and IL-1$\beta$ from ACM-Biotech GmbH).

### Generation of BMDC

BM-derived DC (BMDC) from C57/Bl6 mice were generated from precursor cells as described before (Lutz et al, 1999). In brief, 2 × 106 BMDC per 10-cm dish (BD Falcon) were cultured for 8 d in R10 medium consisting of RPMI1640 (Lonza), 1% penicillin/streptomycin/L-glutamine (Sigma-Aldrich), 2-ME (50 $\mu$M; Sigma-Aldrich), and 10% heat-inactivated FBS (Fetal Bovine Serum Gold; GE Healthcare) and additionally supplemented with GM-CSF supernatant (1:10) from a cell line stably transfected with the murine GM-CSF (Zal et al, 1994). At days 3 and 6, 10 ml of fresh R10 supplemented with GM-CSF supernatant (1:10) was added, by removing 50% of the old cell culture supernatant at day 6 before. Maturation of BMDC was induced at day 8 by the addition of 0.1 ng/ml LPS (Sigma-Aldrich) for 20 h. At day 9, cells were used for further experiments.

## Antibodies

The following antibodies were purchased from Abnova, BD, BioLegend, and Miltenyi and used for immunostaining, flow cytometry, blocking experiments, or immunoblotting of human antigens: CD1c (clone L161), CD1d (51.1), CD11b (ICRF44), CD11c (3.9), CD14 (63D3), CD40 (5C3), CD64 (10.1), CD70 (113-16), CD80 (2D10), CD83 (HB15e), CD86 (IT2.2), CD163 (GHI/61), CD172 (15-414), CD192 (K036C2), CD205 (HD30), CD206 (15-2), CD209 (9E9A8), FCeRI (AER-37(CRA-1)), GM-CSF (BVD2-23B6),

HLA-DR (L243), Slan (M-DC8), Stat5 (py694), and Zbtb46 (H00140685-B01P). The following secondary antibodies and isotype controls were used: anti-mouse IgG (poly4060) and Rat IgG2a (RTK2758). For mouse antigens, see the MELC Antibodies section. Primary antibodies were used at 1–2 $\mu$g·ml$^{-1}$ for immunoblotting, 2 $\mu$g·ml$^{-1}$ for immunofluorescence, and 5–10 $\mu$g·ml$^{-1}$ for blocking experiments and flow cytometric analysis.

## Cell assays

### Mixed lymphocyte reaction

CFSE-labeled T cells were seeded in round bottom 96-well plates in triplicate cultures at 1 × 10$^5$ T cells/well, and indicated ratios of DC were subsequently added for 5–6 d.

### T-cell proliferation assay

To measure proliferation, cells were harvested and CFSE dilution was determined in T cells using flow cytometry. Analysis and calculation of proliferation index was done with the software FlowJo v10.

### Stat5 signaling analysis

To measure Stat5 phosphorylation, isolated 1 × 10$^6$ monocytes were treated for 15 min with 20-$\mu$g EV or as a control with recombinant GM-CSF. Cells were fixed and stained with anti-Stat5 or the corresponding isotype control as recommended by the manufacturer's (BD Phosflow) instructions, and cells were analyzed with flow cytometry.

### Monocyte cell proliferation assay

CFSE-labeled PBMCs were treated with EV or GM-CSF, respectively, for 1 or 10 d. Monocyte proliferation was determined by CFSE dilution in CD11b$^+$-gated cells with flow cytometry. Analysis and calculation of proliferation index was done with the software FlowJo v10.

### Monocyte stimulation

10-$\mu$g EV pellet corresponded to the production of 4 × 10$^6$ maDC or imDC in 24 h (see the Isolation and purification of EV section). This amount was used to stimulate 250,000 monocytes in 1.25 ml once for 6 d. Hence, the in vitro production of 16 maDC in 24 h was sufficient (but not necessarily required as this was not titrated down) to stimulate one monocyte in 6 d. The CCF concentration in 10-$\mu$g EV preparation and in comparison with the factors secreted into the supernatant is shown in Fig S2. For control, 10 $\mu$g of EV purified from the supernatant of 293T cells was used, similarly as described in our previous publications. These vesicles have no CCF content (Lee et al, 2016).

## Cell analysis

### Flow cytometry analysis (FACS)

Cells were stained with fluorochrome-conjugated antibodies, and flow cytometric analysis was done using a FACS Canto II flow cytometer (BD Bioscience). Data were analyzed with the FCS Express 4 (De Novo Software) or FlowJo V10 software.

### Confocal microscopy

For detection of EV uptake, $5 \times 10^5$ monocytes were isolated as described above and treated for 3 h with 10 µg of EV. Monocytes were adhered to slides and fixed with 4% paraformaldehyde. Slides were repeatedly washed in PBS, dried and mounted with Fluoromount-G (Southern Biotech), and analyzed using a confocal laser-scanning microscope (Laser Scanning System [LSM 510 Meta; Zeiss] based on an inverted microscope [Axiovert 200 M; Zeiss]). All the procedures were performed at room temperature.

## EV depletion of FCS and human serum for cell culture

To assure EV generated from cell culture were not contaminated by outside sources, heat-inactivated FCS and human serum for medium supplementation were depleted of vesicles by ultracentrifugation for 18 h at 110,000 $g$ and 4°C before use.

## Isolation and purification of EV

EV purification was performed essentially as described previously (Muratori et al, 2009; Thery et al, 2009). Briefly, DC supernatants were collected after the last 6 d of culturing monocytes ($70 \times 10^6$ cells in 340-ml medium; only the EV production of 24 h was collected) stimulated with GM-CSF/IL-4 to obtain imDC, or after imDC were stimulated for 24 h with a MC to obtain maDC. The supernatants were centrifuged for 20 min at 2,000 $g$ and 30 min at 10,000 $g$ and ultracentrifuged for 1 h at 100,000 $g$. Pellets were re-suspended in 35-ml PBS and centrifuged at 100,000 $g$ for 1 h. Pellets were re-suspended in 300-µl PBS and considered as EV preparations. The total pellet usually contained 150- to 170-µg protein containing EV produced in 24 h. Hence, 10-µg EV pellet corresponded to the production of $4 \times 10^6$ maDC in 24 h.

For gradient purification, EV were diluted in 2 ml of 2.5 M sucrose and 20 mM Hepes/NaOH, pH 7.4, and a linear sucrose gradient (2–0.25 M sucrose, 20 mM Hepes/NaOH, pH 7.4) was layered on top of the EV suspension or EV were diluted in 500-µl homogenization media (HM) of 0.25 M sucrose, 1 mM EDTA, and 10 mM Tris–HCl and layered on top of linear OptiPrep (Axis Shield) gradient (40–5% OptiPrep, HM). The samples were then centrifuged at 210,000 $g$ for 15 h. Gradient fractions were collected from top down, and the refractive index was determined. Each fraction was diluted in 10-ml PBS and ultracentrifuged for 1 h at 110,000 $g$. Pellets were solubilized in SDS sample buffer or re-suspended in 100-µl PBS and analyzed by immunoblotting or CCF protein array (see Human CCF array section).

For labeling of EV with PKH, we used the Sigma Mini26-1KT" PKH26 Red Fluorescent Cell Linker Mini kit (Sigma-Aldrich) according to the manufacturer's procedures.

## FACS analysis of DC-derived EV

The method was performed essentially as described previously (Lee et al, 2016). Latex beads were coated with 10-µg EV preparation as described above. Subsequently, the beads were incubated with anti-GM-CSF in 50-µl PBS/0.5% BSA for 30 min at 4°C. 200-µl PBS/0.5% BSA was added, and the sample was centrifuged at 1,500 $g$ for 3 min at RT. The pellet was re-suspended in 200-µl PBS/0.5% BSA and incubated with 1-µl anti-mouse Alexa Fluor 488–labeled secondary antibody for 30 min at 4°C and subsequently washed twice before a FACS measurement was carried out.

## Mouse injection experiments

### Mice

All experiments were performed in accordance with the European Communities Council Directive (86/609/EEC) and were approved by the local ethics committee (Government of Middle Franconia, Germany). C57BL/6 mice were bred inhouse and kept on a 12-h light/12-h dark cycle and had free access to food and water. Age-matched 12-wk-old mice were used for all experiments.

### Biodistribution in healthy mice

Groups of four mice underwent subcutaneous injections of ma-/imBMDC-EV in 100-µl PBS, equivalent to a total amount of 50-µg EV per mouse. Control mice were injected with either 100-µl PKH26-containing media or PBS alone. After 6 h and 24 h, the mice were anesthetized with isoflurane and euthanized, 1-cm$^2$ skin patches around injection site and draining lymph nodes were harvested, photos were taken, and tissue was embedded in Tissue-Tek O.C.T. and frozen at –80°C until samples were proceeded by MELC technology.

## MELC technology

### MELC sample preparation

Tissue sections of 5 µm were prepared using a cryotome (Leica CM3050 S; Leica), incubated in Aceton (Carl Roth) for 10 min at –20°C, and air-dried for 5 min. For rehydration, the slides were placed in PBS (PAA) for 5 min at RT, followed by incubation with 5% NGS (Dako) in PBS for 30 min in order to block unspecific binding sites.

### MELC antibodies

For MELC analyses, the following fluorophore-labeled antibodies and propidium iodide (Genaxxon Bioscience) were used: anti-B220 (RA3-6B2; BD Pharmingen), anti-CD3e (145-2C11; BD Pharmingen), anti-CD11b (M1/70; BD Pharmingen), anti-CD44 (IM7; BD Pharmingen), anti-CD45 (30-F11; BD Pharmingen), anti-CD54 (3E2; BD Pharmingen), anti-CD64 (X54-5/7.1; BioLegend), anti-CD83 (Michel-19; BD Pharmingen), anti-CD86 (GL1; BD Pharmingen), anti-CD169 (3D6.112; BioLegend), anti-CD206 (C068C2; BioLegend), anti-cytokeratin-14 (LL002; Abcam), anti-F4-80 (CI:A3-1; eBioscience), anti-Ly6C (HK1.4; BioLegend), anti-Ly6G (1A8; BioLegend), and anti-MHC class II (M5/114.15.2; BioLegend). The best working dilutions of the antibodies for the MELC analysis were determined in initial calibration runs, adjusted if necessary, and tested again.

### MELC data generation

The MELC technology has been described previously (Schubert et al, 2006). The coverslip with the sample was positioned onto a motor-controlled XY stage of an inverted fluorescent microscope (Leica DM IRE2; Leica Microsystems; ×20 air lens; numerical aperture, 0.7). The repetitive cyclic process of this method includes the following steps: (a) antigen tagging by a fluorescence-coupled monoclonal

antibody, (b) washing, (c) image assessment, and (d) photo bleaching. By means of a pipetting robot unit, the antibodies were incubated with the sample for 30 min and subsequently rinsed with PBS. Phase contrast and fluorescence signal were assessed by a cooled CCD camera (Apogee KX4; Apogee Instruments; 2048). The photo bleaching step at the excitation wavelengths was connected downstream the washing steps. After completion of the cycle, the next antibody was added to the same tissue sample. Two to four visual fields were recorded simultaneously during each MELC run. Data acquisition was achieved using imaging software developed by the former company MelTec GmbH. For quantification and calculation of the signal intensity, the ROI (regions of interest) manager tool of the ImageJ software was used.

### MELC data analysis

Using the corresponding phase-contrast images, fluorescence images produced after each antibody stain were aligned pixelwise and were corrected for illumination faults using flat-field correction. The alignment reached a resolution of ±1 pixel. Post-bleaching images were subtracted from the following fluorescence tag images. Superimposed images composed a n epitope expression in relation to each pixel ($900 \times 900$-$nm^2$ area) of a visual field ($1{,}024 \times 1{,}024$ pixels).

Protein expression quantification was conducted by StrataQuest software (TissueGnostics) and is described in Fig S5.

### Statistical analysis

Data were statistically evaluated using the $t$ test or one-way ANOVA with Excel or GraphPad Prism software.

## Supplementary Information

## Acknowledgements

This work was supported by funds from the Interdisziplinäre Zentrum für Klinische Forschung (IZKF) of the University of Erlangen/Nürnberg, from the German Science Foundation (DFG) (SFB 643), and from the German Federal Ministry of Education and Research (BMBF) under grant 01GU1107A. S Schierer is supported by the IZKF Erlangen; C Ostalecki by the Comprehensive Cancer Center Erlangen; and E Zinser and L Stich from CRC1181. We are grateful to TissueGnostics for providing the StrataQuest Software.

### Author Contributions

S Schierer: data curation, formal analysis, validation, investigation, and methodology.
C Ostalecki: data curation, formal analysis, investigation, and methodology.
E Zinser: data curation.
R Lamprecht: data curation.
B Plosnita: software.
L Stich: data curation.
J Dörrie: writing—original draft.
MB Lutz: writing—original draft.
G Schuler: funding acquisition and project administration.
AS Baur: conceptualization, resources, formal analysis, supervision, funding acquisition, investigation, project administration, and writing—original draft, review, and editing.

### Conflict of Interest Statement

The authors declare that they have no conflict of interest.

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
