## [Reviewer comments · Life Science Alliance]

Life Science Alliance

Extracellular Vesicles from mature human DC differentiate Monocytes into immature DC

Andreas Baur, Stephan Schierer, Christian Ostalecki, Elisabeth Zinser, Ricarda Lamprecht, Bianca Plosnita, Lena Stich, Jan Doerrie, Manfred Lutz, and Gerold Schuler

DOI: 10.26508/lsa.201800093

Corresponding author(s): Andreas Baur, University of Erlangen-Nürnberg

Review Timeline:

Submission Date:	2018-05-21
Editorial Decision:	2018-06-19
Revision Received:	2018-09-28
Editorial Decision:	2018-10-29
Revision Received:	2018-11-15
Accepted:	2018-11-20

Scientific Editor: Andrea Leibfried

Transaction Report:

June 19, 2018

Re: Life Science Alliance manuscript #LSA-2018-00093-T

Prof. Andreas Stephan Baur
University of Erlangen-Nürnberg
Department of Dermatology
Hartmannstrasse 14
Erlangen, Bavaria 91052
GERMANY

Dear Dr. Baur,

Thank you for submitting your manuscript entitled "Extracellular Vesicles from mature human DC differentiate Monocytes into immature DC and activate murine Ly6C+ Monocytes in tissue" to Life Science Alliance. The manuscript was assessed by expert reviewers, whose comments are appended to this letter. We invite you to submit a revision if you can address the reviewers' key concerns, as outlined here.

As you will see, the referees are currently not convinced that your data support your conclusions. While absence of further reaching in vivo evidence is not a concern precluding publication here, the additional controls (reviewer #1) and repetition of experiments for a better characterization of the immune cells (reviewer #2) need to be addressed for a better understanding of the effects of extracellular vesicles on monocytes, and to better support the conclusions you put forward. As blocking EV secretion in DCs is not an easy task, this request of reviewer #3 does not need to be experimentally addressed. Overall, the revision will require a lot of work and effort, and we will need strong support on it from reviewer #1 and #2. So please consider your options carefully.

-- High-resolution figure, supplementary figure and video files uploaded as individual files: See our detailed guidelines for preparing your production-ready images, <http://life-science-alliance.org/authorguide>

B. MANUSCRIPT ORGANIZATION AND FORMATTING:

Full guidelines are available on our Instructions for Authors page, <http://life-science-alliance.org/authorguide>

Thank you for this interesting contribution to Life Science Alliance. We are looking forward to receiving your revised manuscript.

Sincerely,

Reviewer #1 (Comments to the Authors (Required)):

The article by Schierer et al describes the secretion of extracellular vesicles (EVs) associated with a wide array of cytokines and chemokines by immature and mature dendritic cells, of both human and mouse origins. The amount of some of these molecules is different between immature and mature DC-derived EVs, and functional consequences, on the differentiation of human monocytes in vitro and attraction of murine myeloid cells in vivo are explored. The authors show some specific effects of mature DC-EVs, which induce or attract a particular type of monocyte-derived cells (slan+ human cells). The authors end their paper with a model for secretion of cytokine-associated EVs and their interaction with target cells.

The article is overall interesting, and well performed, although descriptive. Presence of cytokines and/or chemokines in EV preparations has been shown and discussed before (review Buzas et al, Nat rev Rheumatol 2014, primary paper Cossetti et al, Mol cell 2014, and Lee et al EBiomed 2016 = authors of the current article). The novelty of this article is in the effect observed on monocytes differentiation by the DC-derived EVs, which is interesting, although the relevance of these findings to the in vivo situation where DCs release EVs and cytokines/chemokines is difficult to evaluate. What would be the actual contribution of EV-associated versus free molecules normally present in a tissue to monocyte differentiation and recruitment? This is a difficult to answer question, though.

Major points:

1) At least, the authors should show, for some of the major cytokines detected in/on EVs, what is the proportion of EV-associated vs free cytokine released by the cells (by comparing actual amount/ml of conditioned medium before and after EV isolation, and in the EV pellet, of this cytokine), and if possible, demonstrate a differential activity of a cytokine as free versus EV-associated form.

As currently presented, the article does not provide any quantitative information on such secretion. The amount of cytokines recovered in EVs is expressed in pg/ml, but also indicating that it is measured in 50microg of EVs : from how many secreting cells are these 50microg of EVs obtained? How much free cytokine would be quantified in the conditioned medium of this amount of cells? For the functional assays, what is the actual amount of the cytokines provided to monocytes incubated with 10 or 30 microg of EVs ? What would be the effect of a similar amount of soluble, non-EV associated cytokine ?

2) The model of figure 8b is not consistent with knowledge on secretion of EVs : EVs either form by budding from the plasma membrane or as intraluminal vesicles of multivesicular endosomes, which then fuse with the PM to release their content. Here the vesicles are depicted as intracellular vesicles, probably secretory vesicles containing cytokines that have followed the regular endoplasmic reticulum-secretory pathway. These vesicles are shown as released through a break in the PM, instead of fusing themselves with the PM to release their soluble content : such extracellular vesicles would thus derive from broken, dying cells, and display the inverse membrane orientation of regular PM- or MVB-derived EVs (cytosolic side outside instead of inside). I am not sure why the authors favor this model where the cytokine is inside EVs, when they acknowledge that their anti-GM-CSF blocking experiments suggest instead it to be outside... In fact, for a cytokine protected/masked inside a vesicle to act on a target cell requires yet an additional step to release this molecule. Therefore, another model whereby cytokines are regularly secreted but then secondarily bind to specific receptors (eg anti-GM-CSF) or promiscuous receptors (eg proteoglycans) on the surface of EVs would explain much more easily the authors observations. In any case, the authors must actually determine the localization of GM-CSF (or any other cyto/chemokine of their choice) inside or outside EVs: eg by mild trypsin digestion in the presence or not of detergent, followed by WB, or by dot-blot assay plus/minus detergent (as described in

MacKenzie...Weaver Cell rep 2016 (suppl figures) or Lai ...Breakefield Nat Comm 2015), or by flow cytometry after capturing the EVs on beads to facilitate detection by regular flow cytometer.

Minor points:

- 1) Missing information in Mat&Meth on the protocol for electroporation of GFP mRNA in DCs: is it performed on mature or immature DCs, do the authors control level of cell death, and how does this process affect the nature of released EVs?
- 2) Missing information in Mat&Meth on the amount of EVs recovered per cultured immature and mature DC and volume of conditioned medium? What do the doses used for assays (10 microg, 30 microg...) correspond to in terms of donor cells (and thus donor to recipient ratio) ?
- 3) Figure S2B is important to show association of cytokines with EVs, but depending on the way the gradient is handled, separation of soluble and EV-associated components is not as efficient: are the authors performing a bottom-up floatation or a top-to-bottom pelleting? Bottom-up separates floating EVs from particulate non-EV materials that remain at the bottom, whereas vesicles and particulate materials go down in top-to-bottom gradients, while soluble proteins probably do not enter the gradient and remain on top. Furthermore, is the gradient made of sucrose (as indicated on the figure) or iodixanol (as indicated in the legend) ?
- 4) the dendritic cell nature of differentiated monocytes is not very clear: markers shown in fig2 may suggest macrophages rather than DCs, images of figure 1 could correspond to any cells. Performing cytopspin to make the cells spread would be more appropriate to distinguish the morphology of DCs and macrophages.
- 5) Indicate clearly if individual experiments involve different donors of recipient monocytes or different donors of EV-producing monocytes/im/maDCs, and perform each functional experiment with at least two biological replicates of EVs (each possibly on different recipient monocytes).
- 6) Many experiments displayed as representative should show in addition (or instead) all experiments: for instance fig1E (possibly choosing only one DC:T cell ratio), figure 3D (is the fold increase calculated from the representative image or from the 4 experiments?)...
- 7) Some remaining bar graphs should be replaced (or modified) to indicate the position of individual biological replicates, if possible with different symbols for each individual experiment (fig 2A,E, 3B, 6B). As recommended by: Weissgerber et al, Plos Biol 2015, 13(4): e1002128.
- 8) Two recent papers analysing in great details EVs from human DCs in terms of composition (Kowal...Thery PNAS 2016) and T cell activation ability (Tkach...Thery EMBO J 2017) should be at least discussed here in terms of consistency or not with the here displayed results. For instance, if cyto/chemokines were not identified by the proteomics data of the other group, could it be because the levels in EVs are too low to be detectable by proteomics? Also these previous articles show that different types of EVs are released by DCs, and that they display some common functions, and some specific. In the current article, the authors analyse only a pellet of small EVs (corresponding to 100K of the other authors): whether the cyto/chemokine-bearing vesicles are mainly this subpopulation of EVs, or whether larger ones could also contribute, since they are most likely also released in vivo, should also be discussed.

Reviewer #2 (Comments to the Authors (Required)):

The current manuscript by Schierer et al. describes a possible role for DC-derived extracellular vesicles (EV) in the differentiation of monocytes into inflammatory DCs.

The authors found that EV derived from human mature DCs induce the differentiation of monocytes into immature DCs. Similar findings were obtained in the murine system using EV obtained from BMDCs. Furthermore the injection of EV derived from mature BMDCs into mouse skin

led to the recruitment and activation of monocytes.

In general the paper is written in a rather confusing and complicated manner and together with the large volume of figures makes it difficult to read and to understand the major findings. This is especially true for the mouse part of the manuscript, which is very convoluted and raises concerns about the correct interpretation of the data. This *in vivo* part including the description of the infiltrating cells is not very well characterized. In addition, the mechanism of how EV induce the accumulation and 'differentiation/activation' of monocytes *in vivo* is not clear. Thus, the main conclusion that EVs activate murine Ly6C⁺ monocytes in tissue is not fully supported by their data.

- Could the authors not use multi-parameter flow cytometry at different time points after the EV injections for a better characterization of the immune infiltrates?

They employ a technique for the detection of multiple surface antigens in tissue sections, which gives them the ability to identify immune cell types based on the co-expression of different markers, which is essential in order to correctly distinguish different myeloid cell types. Nevertheless, the authors often use single markers to identify a cell population, which somehow misses the point of using this technique. For example, the authors refer to Ly6C⁺ cells as monocytes, even though Ly6C is expressed on other cell types such as neutrophils and T cells. Other monocyte-derived markers, for example CD11b, should be used together with Ly-6C to correctly identify monocytes, and together with absence of Ly6G immunoreactivity. In Figure 5A, last row, one can clearly see an area where cells are positive for Ly6C, but not CD11b, but the authors describe these cells as monocytes, and refer to CD11b as only a differentiation marker for monocytes.

- This becomes even more confusing as often the same dataset is represented in different ways - frequency of cells based on the expression of one marker or frequency of cells based on a combination of markers - which leads to one cell subset being referred to differently in different contexts. Furthermore, the authors avoid using the cell type names for some of these combinations. For example CD45⁺Ly6G⁺Ly6C⁺ should be referred to as neutrophils to make it easier to understand and follow the text.

- The quantification of the different marker combinations, especially in the time point of 6 hours after injection, is also difficult to understand. In Table 1 the authors provide the numbers of cells positive for a given combination of markers within the EV-area. In the first column, one can see that 63 out of 68 total cells (92.7%) are CD45⁺ but 65 out of 68 total cells are CD45⁺Ly6C⁺Ly6G⁺. How is it possible that there are more triple positive cells than CD45⁺, when the CD45⁺Ly6C⁺Ly6G⁺ should be a subset of all CD45⁺ cells? In the other columns however, the reported frequency of cells with combination of markers is then the frequency of CD45⁺ cells instead of total cells. There is clearly a mistake in the table, which should be corrected.

- Following this quantification, the authors state: "Cells expressing CD45 alone, or triple marker combinations of the most abundant markers (CD45/Ly6G/Ly6C and CD45/Ly6C/F4-80) were present in both areas at a similar proportion (84-95%, Table 1, red box)." Since CD45/Ly6G/Ly6C cells would correspond to neutrophils and CD45/Ly6C/F4-80 - to monocytes, this suggests that there are cells, which are positive for all 4 markers, or that the total frequency of cells is more than 100%. However, Ly6G and F4/80 are exclusive markers for different cell types, so this questions the validity of the method and the quantification. The same discrepancy can be seen in other panels (Figure 4B, 6C, etc). Figure 4A, lower graph, for example implies that ~80% of cells are monocytes, ~80% of cells are neutrophils and so on, which would add up to more than 100%. Finally, in the results text the authors seem to imply that they are dealing with a change of cell phenotype ("However, most of the cells had changed from a Ly6C/Ly6G double positive to a Ly6C/F4-80

positive phenotype") rather than different cell infiltrates at different times. The wording of the above sentence seems to suggest that Ly6C/Ly6G cells (neutrophils) transition to Ly6C/F4-80 cells (monocytes). Based on this it is impossible to identify from the text whether the cells that are further defined as monocyte-derived DCs are actually derived from monocytes. The entire section describing these results should be rewritten, rearranged and revised to avoid scientific inaccuracies and to make it clearer.

- The microscopy pictures provided are of very low size, making it difficult to assess whether a particular staining is real. While zoomed in pictures of individual cells are provided (Figure 6A), they are of very low quality. Therefore, the authors should provide high-quality confocal microscopy pictures that show the co-localization of the most common marker combinations used (CD45-Ly6C-Ly6G, CD45-Ly6C-F4/80).
- In addition, are the injected vesicles ingested by monocytes? Can they be detected inside the cells? Confocal microscopy should also be used here.
- The authors identify differentiation markers being expressed on monocytes after EV injection. These results should also be shown by FACS staining to confirm and validate the microscopy technique.
- The authors state that they don't observe the upregulation of certain markers (CD11c, CD206, CD64) on monocytes and explain it with the short duration of the experiments. Therefore, these experiments should be repeated with a longer timeframe to observe whether the aforementioned markers would become expressed.
- Ly6C downregulation has also been reported as a result of monocyte differentiation to moDC. Therefore, the authors should add experiments in which monocytes are labeled prior to EV injection, and follow them, in order to find out whether EV can also induce Ly6C downregulation in the course of monocyte differentiation.
- In the in vitro part of the study, the authors identify GM-CSF as a factor important for the EV-driven differentiation of monocytes. These observations should be repeated in the in vivo model, using EV derived from GM-CSF-deficient mice to show whether the same mechanism applies.
- Fig. 2A: Please show representative FACS plots for the markers. Are the markers uniformly expressed on all cells or are there distinct subpopulations within the differentiated monocytes?

Minor points:

- Fig. 1E: what are control EV?
- Fig. 2A: many of the shown markers are not mentioned at all in the result section. Please consider moving those to the supplementary figure or briefly mentioning them in the text to facilitate the reading and interpretation of the figure.
- Fig. 2A and B: the plots could be rearranged according to the order in which they appear in the text or at least grouped according to the text.
- Fig. 2B: IL-1b is not mentioned at all in the results section for this figure. Please either remove it or

discuss it.

- Table 1: the color of the text under the microscopy pictures and the color of the highlighted boxes makes the figure confusing as it seems to imply a connection between the text and the boxes. Please use different colors or no colors.

- Fig. 3B, why is the effect of anti-GM-CSF antibody less pronounced in the EVimDC compared to EVmaDC? In addition, the authors write in the results "Monocytes treated with imDC-EV, maDC-EV or recombinant GM-CSF revealed a strong Stat5 tyrosine phosphorylation as compared to non-stimulated cells. This effect was blocked in the presence of an anti-GM-CSF antibody (Figure 3B, red box)." However, the mentioned figure does not show Stat5 phosphorylation but the % of FSChigh CD11b+ cells. The authors should fix the text or include the correct figure. The reduction of Stat5 phosphorylation could be also added to Figure 3A.

- Fig. 3C/D. There is a big difference in the levels of GM-CSF in EVimDC compared to EVmaDC in Fig. 3D, which can not be observed in Fig. 3B. Is there an explanation for this?

- Fig. 3D: "fold in. im/ma" should be the other way around (fold increase ma/im)

- All figures are labeled with Schierer et al... whereas tables are labeled as Ostalecki et al...

Reviewer #3 (Comments to the Authors (Required)):

This manuscript provides some intriguing data with regard to the potential mechanisms of immature DC differentiation by extracellular vesicles (EV). Although the functional phenotype looks interesting, the mechanistic support for the involvement of EV would need to be further solidified.

1). The key cytokine (or other soluble mediators) within the EV or released by EV responsible for the observed effect is not identified.

2) To further build a compelling causative connection, approaches that can block the EV release should be applied to perform the functional studies.

We would like to thank the referees for their constructive comments. We tried to address all of them and changed the manuscript considerably, and we think we made it a better publication.

Reviewer #1 (Comments to the Authors (Required)):

Major points:

1) At least, the authors should show, for some of the major cytokines detected in/on EVs, what is the proportion of EV-associated vs free cytokine released by the cells (by comparing actual amount/ml of conditioned medium before and after EV isolation, and in the EV pellet, of this cytokine), and if possible, demonstrate a differential activity of a cytokine as free versus EV-associated form.

*To measure cytokines produced by mature DC, secreted either by EV or directly into the supernatant, we analyzed the production of 70 million maDC cultured in 340 mL of medium for 24 hours. Cytokines were assessed in the supernatant and the purified EV pellet by multiplex/FACS analysis using a commercial assay from BioLegend. We could not analyze GM-CSF, TNF and IL-6 as these cytokines were used to produce and stimulate the cells in vitro. Their values in the supernatant were much higher than all the other cytokines and hence were not incorporated into the analysis. Instead we analyzed IL-10, IL-21 and IFN γ . The results demonstrate that some cytokines (IL-10, IL 21) are mainly secreted directly into the supernatant, while only a small proportion is secreted through the vesicle secretion system. The differences were between 1000-6000 fold. Conversely, IFN γ was secreted at the much higher proportion by vesicles and the difference was only 40 fold. The data are presented in **Figure S2B***

As currently presented, the article does not provide any quantitative information on such secretion. The amount of cytokines recovered in EVs is expressed in pg/ml, but also indicating that it is measured in 50microg of EVs : from how many secreting cells are these 50microg of EVs obtained?

*For harvesting EV, DC were seeded at a concentration of 200.000 cells/ml., e.g. 70 Mio maDC in 350 ml of medium. These cells produced an EV pellet of around 150-170 μ g in 24h. Hence 10 μ g EV pellet corresponded to the production of 4 mio. maDC in 24h. This amount was used to stimulate 250.000 monocytes in 1.25 ml, meaning that the in vitro production of 16 maDC in 24h was sufficient (but not necessarily required as this was not titrated down) to stimulate 1 monocyte. This is now explained in the main text as well as **M&M** (in Cell assays: monocyte stimulation).*

How much free cytokine would be quantified in the conditioned medium of this amount of cells?

*See explanation above and **Figure S2B**.*

For the functional assays, what is the actual amount of the cytokines provided to monocytes incubated with 10 or 30 microg of EVs ?

*We have now included a table with the amounts of the most cytokines/chemokines measured in 10 μ g EV pellet. The analysis was done on additionally purified EV preparations / EV pellets, see **Figure S2A**.*

What would be the effect of a similar amount of soluble, non-EV associated cytokine ?

*We have not yet incubated monocytes with the same amounts of cytokines found in EV pellets, but we can provide the amounts/concentrations of cytokines necessary to stimulate monocytes in order to generate monocyte-derived im- and maDC: in 6 days 141 ng/ml GM-CSF and 22.7 ng/ml IL-4 is added 3 times. To mature DC on day 6, 2 ng/ml IL1 beta, 6.6 ng/ml IL6, 10 ng/ml TNF and 1 μ g/ml PGE2 is added. Hence, with respect to GM-CSF there is at least about 170fold more soluble GMCSF needed to stimulate cells than EV-associated GM-CSF (about 2,5 ng/10 μ g EV). This is now explained in the **main text**.*

2) The model of figure 8b is not consistent with knowledge on secretion of EVs : EVs either form by budding from the plasma membrane or as intraluminal vesicles of multivesicular endosomes, which then fuse with the PM to release their content. Here the vesicles are depicted as intracellular vesicles, probably secretory vesicles containing cytokines that have followed the regular endoplasmic reticulum-secretory pathway. These vesicles are shown as released through a break in the PM, instead of fusing themselves with the PM to release their soluble content : such extracellular vesicles would thus derive from broken, dying cells, and display the inverse membrane orientation of regular PM- or MVB-derived EVs (cytosolic side outside instead of inside).

I am not sure why the authors favor this model where the cytokine is inside EVs, when they acknowledge that their anti-GM-CSF blocking experiments suggest instead it to be outside... In fact, for a cytokine protected/masked inside a vesicle to act on a target cell requires yet an additional step to release this molecule. Therefore, another model whereby cytokines are regularly secreted but then secondarily bind to specific receptors (eg anti-GM-CSF) or promiscuous receptors (eg proteoglycans) on the surface of EVs would explain much more easily the authors observations.

In our 2013 paper (Lee et al., Supplement Figure 7) we demonstrate how vesicular structures containing mature/processed cytokines (TNF) develop in the cytoplasm. In our 2016 paper (Ostalecki et al.), using double labeled cytokine precursors (GFP-proTNF-RFP), we demonstrate how the cytokine precursors are processed in the cytoplasm (in this case by ADAM17), showing that the mature TNF faces inward in intracellular and secreted vesicles (see Figure 1d below and also Figure 4 in the paper). These vesicles (Rab27+) with mature TNF were transferred directly onto or into neighboring cells (see Figure 1e below). Also, using MELC technology, we demonstrate the transfer of cargo/cytokine containing vesicles into neighboring cells, using the example of melanocytes and keratinocytes (Ostalecki et al., 2017). Together the results suggest that the vesicular transport of TNF from one cell to the other is rather common and effective. Results from my colleagues in Finland suggest that this secretion type is originating directly from the Golgi, and is regulated by tyrosine kinases (see also our paper in 2018 Lee et al.) probably through the so-called non-conventional secretion pathway (paper in revision). In view of all of these data it was reasonable to assume a similar mechanism for GM-CSF.

Ostalecki/Wittki et al. Fig.1

Ostalecki et al. 2016, Figure 1: HIV pEV induce endosomal proTNF cleavage (D) Spatial orientation of G-pTNF-R in endosomes. 293T cells were transfected with G-pTNF-R and analyzed by confocal microscopy after 24 hours. **(E)** HIV pEV induce a vesicular secretion mechanism. G-pTNF-R transfected 293T cells (12h) were incubated with HIV pEV (1 ml plasma equivalent pooled from different donors) for 8h, mixed with non-transfected cells (1:4; 12h) and analyzed by confocal microscopy. **(F)** HIV pEV induce proTNF cleavage in macrophages. Macrophages were incubated (16h) with pEV-aliquots as in (A) before yellow (proTNF) and red (mature TNF) vesicular compartments were quantified in % of total vesicles, counted on one confocal level (examples at the bottom) of 20 randomly selected cells for each condition. Error bars indicate standard deviation of the mean of 20 cells. **(G)** HIV pEV induce TNF release in the not-adherent PBMC fraction (NAF: T /B cells). PBMC and the

NAF fraction of the same PBMC were incubated with HIV pEV as in (A). In addition, cells were stained for TNF by confocal microscopy as indicated. Error bars were calculated on the basis of triplicates.

In any case, the authors must actually determine the localization of GM-CSF (or any other cyto/chemokine of their choice) inside or outside EVs: eg by mild trypsin digestion in the presence or not of detergent, followed by WB, or by dot-blot assay plus/minus detergent (as described in MacKenzie...Weaver Cell rep 2016 (suppl figures) or Lai ...Breakefield Nat Comm 2015), or by flow cytometry after capturing the EVs on beads to facilitate detection by regular flow cytometer.

We have done such an orientation study for ADAM10 and 17 and came to the same conclusion as Cvjetkovic et al. 2016, namely that many proteins in extracellular vesicles have an inward orientation. The result of this study was used to design a specific protease assay where its substrate (a FRET peptide) had to be modified so it would penetrate the EV membrane in order to reach the active center of the protease and be processed physiologically (e.g. Lee et al. 2016).

*For the here presented paper we have stained DC-derived vesicles (coupled to beads) and saw staining with the GM-CSF blocking antibody (now **Figure S4B**) confirming our blocking experiment. Beyond that, we think that the precise orientation of GM-CSF in vesicles is not the main focus in this paper and was addressed by us and others previously. Our model was presented as part of the discussion. We have made clear, that this is only a model and stated that other explanations are possible.*

Minor points:

1) Missing information in Mat&Meth on the protocol for electroporation of GFP mRNA in DCs: is it performed on mature or immature DCs, do the authors control level of cell death, and how does this process affect the nature of released EVs?

Concerning the electroporation, we have cited the work (from our department) describing in detail this procedure. Only for one experiment we used GFP labeled EV, namely for the very first experiment (Figure 1 A) in order to demonstrate the uptake of vesicles. Electroporation by RNA does not overly stress the cells and is part of a standard procedure in our Department to generate a cancer vaccine. In order to mature cells, the monocytes are cultured at first for 6 days with GM CSF and IL-4, before the cell are washed, removing remaining debris from dying cells, and the maturation cocktail is added.

For all other experiments, the vesicles were stained with PKH26 after purification. Such labeled EV have also been used to differentiate cells and no difference to non-stained vesicles was noted.

2) Missing information in Mat&Meth on the amount of EVs recovered per cultured immature and mature DC and volume of conditioned medium? What do the doses used for assays (10microg, 30 microg...) correspond to in terms of donor cells (and thus donor to recipient ratio) ?

See explanation above.

3) Figure S2B is important to show association of cytokines with EVs, but depending on the way the gradient is handled, separation of soluble and EV-associated components is not as efficient: are the authors performing a bottom-up floatation or a top-to-bottom pelleting? Bottom-up separates floating EVs from particulate non-EV materials that remain at the bottom., whereas vesicles and particulate materials go down into top-to-bottom gradients, while soluble proteins probably do not enter the gradient

and remain on top. Furthermore, is the gradient made of sucrose (as indicated on the figure) or iodixanol (as indicated in the legend) ?

*The Gradients were made with iodixanol (Optiprep) and top to the bottom pelleting, which is now detailed in the **M&M** section, (in: Isolation and purification of EV)*

4) the dendritic cell nature of differentiated monocytes is not very clear: markers shown in fig2 may suggest macrophages rather than DCs, images of figure 1 could correspond to any cells. Performing cytopspin to make the cells spread would be more appropriate to distinguish the morphology of DCs and macrophages.

I understand the referee`s criticism, however in the DC field cells are usually shown this way in order to demonstrate the veils that these cells characteristically develop. In our case the development of a cell with a regular roundish membrane into a cell with many veils emanating from the membrane is the main purpose of the figure panel. If these cells were cytopspined, they would adopt a morphology that is very difficult to discriminate from monocytes or immature dendritic cells.

5) Indicate clearly if individual experiments involve different donors of recipient monocytes or different donors of EV-producing monocytes/im/maDCs, and perform each functional experiment with at least two biological replicates of EVs (each possibly on different recipient monocytes).

We have added this information when missing. In general all experiments were performed at least 5 to 10 times with different donors and acceptors over a time span of several years, in which the project was completed.

6) Many experiments displayed as representative should show in addition (or instead) all experiments: for instance fig1E (possibly choosing only one DC:T cell ratio)

*This has been done now in **Figure S3A**.*

and figure 3D (is the fold increase calculated from the representative image or from the 4 experiments?)

*The fold increase displayed was calculated for the respective data set and not for all four experiments (we performed more than 4 experiments). A second example is now shown in **Figure S4C**. Unfortunately the protein array from RayBiotech showed considerable variability, even though the general trend was the same. This may have been due to donor variability, operator handling and the extremely sensitive HRP-based enzymatic detection system. This was one of the reasons we employed a second system, namely the FACS-based multiplex system from BioLegend, which we repeated with many donors. For the CCF assessment in Figure 5D-E the multiplex system is more representative, as it shows less variability and is more accurate in determining the CCF concentrations.*

7) Some remaining bar graphs should be replaced (or modified) to indicate the position of individual biological replicates, if possible with different symbols for each individual experiment (fig 2A,E, 3B, 6B).

As recommended by: Weissgerber et al, Plos Biol 2015, 13(4): e1002128.

By adding all the additional data in the supplement material we hope to have satisfied the referees request for more representative data.

8) Two recent papers analysing in great details EVs from human DCs in terms of composition (Kowal...They PNAS 2016) and T cell activation ability (Tkach...They EMBO J 2017) should be at least

discussed here in terms of consistency or not with the here displayed results. For instance, if cyto/chemokines were not identified by the proteomics data of the other group, could it be because the levels in EVs are too low to be detectable by proteomics? Also these previous articles show that different types of EVs are released by DCs, and that they display some common functions, and some specific. In the current article, the authors analyse only a pellet of small EVs (corresponding to 100K of the other authors): whether the cyto/chemokine-bearing vesicles are mainly this subpopulation of EVs, or whether larger ones could also contribute, since they are most likely also released in vivo, should also be discussed.

We have now cited the two papers and discussed the issue of CCF detection raised by the referee. As far as we can see the list of proteins analyzed by proteomics are bigger than 10 kDa (see list in supplement data of Tkach et al.), most cytokines, however, have a size of 8 to 10 kDa. It is therefore possible that in this paper the here reported CCF factors were not detected. Despite being a powerful technique, application of mass spectrometry to cytokine analysis is challenging. Several limitations of MS should be considered: (1) MS is able to detect only charged (ionised) peptides but not all peptides are ionised with the same efficiency. Moreover, proteins with low molecular weight (including most cytokine molecules) provide less peptides and taken together with low abundance, these peptides will not be detected without enrichment or prefractionation. (2) Sensitivity of shotgun MS is usually insufficient for detection of low abundant proteins (such as cytokines) in complex biological samples without employment of sample depletion or prefractionation.

The vesicles secreted by DC usually have a size of about 120nm +/- 20nm as determined by the ZetaView® technology and are similar to the vesicles we described in our previous publications. We think that the main focus of this paper is to demonstrate that DC-derived EV can differentiate monocytes. The precise size and nature of these vesicles may be described in a separate study.

Reviewer #2 (Comments to the Authors (Required)):

In general the paper is written in a rather confusing and complicated manner and together with the large volume of figures makes it difficult to read and to understand the major findings. This is especially true for the mouse part of the manuscript, which is very convoluted and raises concerns about the correct interpretation of the data. This in vivo part including the description of the infiltrating cells is not very well characterized. In addition, the mechanism of how EV induce the accumulation and 'differentiation/activation' of monocytes in vivo is not clear. Thus, the main conclusion that EVs activate murine Ly6C+ monocytes in tissue is not fully supported by their data.

The main message of the paper is the demonstration that vesicles from mature DC induce differentiation of monocytes towards inflammatory dendritic cells. Proving a detailed mechanism in vivo is not that easy and likely requires an additional project. I'd also like to point out that we were careful not to make bold statements regarding the activation of Ly6C cells or monocytes and rather stated that these cells developed activation markers. In the extensively revised manuscript we tried to address all the issues raised by the referee.

- Could the authors not use multi-parameter flow cytometry at different time points after the EV injections for a better characterization of the immune infiltrates?

We feel that the topographical analysis of immune infiltration by the MELC technology brings the decisive insight into the effect of vesicles. I guess we would lose this advantage using multi parameter flow cytometry.

They employ a technique for the detection of multiple surface antigens in tissue sections, which gives them the ability to identify immune cell types based on the co-expression of different markers, which is essential in order to correctly distinguish different myeloid cell types. Nevertheless, the authors often use single markers to identify a cell population, which somehow misses the point of using this technique. For example, the authors refer to Ly6C⁺ cells as monocytes, even though Ly6C is expressed on other cell types such as neutrophils and T cells. Other monocyte-derived markers, for example CD11b, should be used together with Ly-6C to correctly identify monocytes, and together with absence of Ly6G immunoreactivity. In Figure 5A, last row, one can clearly see an area where cells are positive for Ly6C, but not CD11b, but the authors describe these cells as monocytes, and refer to CD11b as only a differentiation marker for monocytes.

The referee is correct. While I usually was careful to be purely descriptive in this respect and avoid incorrect conclusions, e.g. using the term Ly6C positive cells rather than Ly6C monocytes, I found two examples in the text where I wrote Ly6C monocytes. This was corrected, also in Figure 4.

*In addition, in Fig. 5B we had already quantified the Ly6C/CD11b-pos. cells (ca 35%). Following the suggestion of the referee, we have concentrated on the quantification and analysis of monocytes and neutrophils (new panel **Figure 4C**) and added new calculations in Figure 5B showing CD11b⁺/Ly6C⁺/Ly6G⁻ cells (true monocytes) with all additional markers (each separately).*

The single marker depiction in Figure 4 and 5 was displayed in order to demonstrate the overall distribution and presence of these markers in areas injected with vesicles of different origin (from immature or mature DC). The novelty here is that all these markers are displayed on one single tissue section. To the best of my knowledge this is the first time that these many markers have been analyzed on one tissue section and in combinations. In Figure 6 we then wanted analyzed combinations of markers to assess and characterize potential differentiation or activation events.

- This becomes even more confusing as often the same dataset is represented in different ways - frequency of cells based on the expression of one marker or frequency of cells based on a combination of markers - which leads to one cell subset being referred to differently in different contexts. Furthermore, the authors avoid using the cell type names for some of these combinations. For example CD45⁺Ly6G⁺Ly6C⁺ should be referred to as neutrophils to make it easier to understand and follow the text.

As explained above I intended to be foremost descriptive when describing cells with different marker combinations, rather than be decisive in identifying a cell type in order to avoid misinterpretations. One of the reasons is that in densely packed tissue areas the allocation of a combination of markers to one cell is not always as easy and sharp as it is possible in FACS analyses. We explained this now in the main text. Furthermore, we saw marker combinations that are not easily identified as specific subsets (e.g. in Figure 6A) and may represent differentiating cells. In order to solve the situation, we would like to remain descriptive in that sense and concentrate on general shifts of marker expressions in EV tissue areas (Figure 6) rather than individual marker combinations on one cell. Therefore we removed our calculations on triple positive myeloid cells (e.g. CD45⁺/Ly6C⁺/F4-80) and concentrated on monocytes (CD11b⁺/Ly6C⁺/Ly6G⁻) and neutrophils (CD45⁺/Ly6C⁺/Ly6G⁺). Furthermore we recalculated all numbers

setting a slightly higher threshold value for all markers, which lowered the number of positive cells for a given marker.

- The quantification of the different marker combinations, especially in the time point of 6 hours after injection, is also difficult to understand. In Table 1 the authors provide the numbers of cells positive for a given combination of markers within the EV-area. In the first column, one can see that 63 out of 68 total cells (92.7%) are CD45+ but 65 out of 68 total cells are CD45+Ly6C+Ly6G+. How is it possible that there are more triple positive cells than CD45+, when the CD45+Ly6C+Ly6G+ should be a subset of all CD45+ cells? In the other columns however, the reported frequency of cells with combination of markers is then the frequency of CD45+ cells instead of total cells. There is clearly a mistake in the table, which should be corrected.

The referee is correct. In this table there were mistakes in the number calculations that have been corrected. Now all numbers in % refer to the total number of cells found in the EV area. In addition, we removed our calculations on triple positive myeloid cells (e.g. CD45/Ly6C/F4-80) (except for Figure 6C-D) and concentrated on monocytes and neutrophils.

- Following this quantification, the authors state: "Cells expressing CD45 alone, or triple marker combinations of the most abundant markers (CD45/Ly6G/Ly6C and CD45/Ly6C/F4-80) were present in both areas at a similar proportion (84-95%, Table 1, red box)." Since CD45/Ly6G/Ly6C cells would correspond to neutrophils and CD45/Ly6C/F4-80 - to monocytes, this suggests that there are cells, which are positive for all 4 markers, or that the total frequency of cells is more than 100%. However, Ly6G and F4/80 are exclusive markers for different cell types, so this questions the validity of the method and the quantification. The same discrepancy can be seen in other panels (Figure 4B, 6C, etc). Figure 4A, lower graph, for example implies that ~80% of cells are monocytes, ~80% of cells are neutrophils and so on, which would add up to more than 100%.

We know that the thresholds we set for individual markers were low, hence resulting in a higher range of positivity for most markers that may not reflect a specific phenotype of a cell. The rationale for this approach was the assumption that these cell are probably in a process of differentiation and/or state of transition either acquiring or losing a markers. The idea was to assess a shift of these markers in different areas where EV had been injected reflecting these differentiation processes as a whole without sharply discriminating individual cell populations. We now have decreased these threshold values (see explanation above).

Finally, in the results text the authors seem to imply that they are dealing with a change of cell phenotype ("However, most of the cells had changed from a Ly6C/Ly6G double positive to a Ly6C/F4-80 positive phenotype") rather than different cell infiltrates at different times. The wording of the above sentence seems to suggest that Ly6C/Ly6G cells (neutrophils) transition to Ly6C/F4-80 cells (monocytes). Based on this it is impossible to identify from the text whether the cells that are further defined as monocyte-derived DCs are actually derived from monocytes. The entire section describing these results should be rewritten, rearranged and revised to avoid scientific inaccuracies and to make it clearer.

Although we did not want to imply such a transition, the transformation of neutrophils to monocytic cells and into DC has been described in vitro and in vivo, notably in inflammatory lesions (Oehler et al., JExMed 1998). Following the suggestions of the referee we have greatly rephrased and clarified this section, for example removing our analysis on triple positive cells.

- The microscopy pictures provided are of very low size, making it difficult to assess whether a particular staining is real. While zoomed in pictures of individual cells are provided (Figure 6A), they are of very low quality. Therefore, the authors should provide high-quality confocal microscopy pictures that show the co-localization of the most common marker combinations used (CD45-Ly6C-Ly6G, CD45-Ly6C-F4/80).

We now provide these confocal microscopy images showing the colocalization of the common marker combinations including monocytes as requested by the referee (Figure S8B and S11). We would like to point out, however, that confocal images from tissues do not have the same quality as from in vitro cultivated of cells.

- In addition, are the injected vesicles ingested by monocytes? Can they be detected inside the cells? Confocal microscopy should also be used here.

Also this is demonstrated now by new confocal images (Figure S8B).

- The authors identify differentiation markers being expressed on monocytes after EV injection. These results should also be shown by FACS staining to confirm and validate the microscopy technique. *In this case I'd like to refer to our in vitro data, where we analyzed marker expression in vitro in detail.*

- The authors state that they don't observe the upregulation of certain markers (CD11c, CD206, CD64) on monocytes and explain it with the short duration of the experiments. Therefore, these experiments should be repeated with a longer timeframe to observe whether the aforementioned markers would become expressed.

We have tried that, however, after 24 hours labeled DC vesicles rapidly disappear and we could not detect sufficient amount of these structures in order to obtain meaningful conclusions.

- Ly6C downregulation has also been reported as a result of monocyte differentiation to moDC. Therefore, the authors should add experiments in which monocytes are labeled prior to EV injection, and follow them, in order to find out whether EV can also induce Ly6C downregulation in the course of monocyte.differentiation.

- In the in vitro part of the study, the authors identify GM-CSF as a factor important for the EV-driven differentiation of monocytes. These observations should be repeated in the in vivo model, using EV derived from GM-CSF-deficient mice to show whether the same mechanism applies.

These are excellent experiments, however, technically challenging and require a whole series of in vivo experiments. At present we feel that this is beyond the scope of this paper. In this first publication we want to publish the principal observation, namely that DC vesicles induce differentiation of monocytes.

- Fig. 2A: Please show representative FACS plots for the markers. Are the markers uniformly expressed on all cells or are there distinct subpopulations within the differentiated monocytes?

These FACS plots have been provided now (Figure S3B).

Minor points:

- Fig. 1E: what are control EV?

*Controlling EV were purified from the supernatant of 293 T cells. These EV do not contain cytokines or chemokine found in EV of a DC. We have also used pEV (plasma extracellular vesicles) from healthy individuals (not in experiments that are shown). Also these vesicles had no effect on monocytes. We have added this information to the M&M section and the legend of **Figure 3A**.*

- Fig. 2A: many of the shown markers are not mentioned at all in the result section. Please consider moving those to the supplementary figure or briefly mentioning them in the text to facilitate the reading and interpretation of the figure.

We have followed the referee`s advice and described the markers in the text.

- Fig. 2A and B: the plots could be rearranged according to the order in which they appear in the text or at least grouped according to the text.

We have followed the referee`s advice and reordered the markers.

- Fig. 2B: IL-1b is not mentioned at all in the results section for this figure. Please either remove it or discuss it.

This marker is now described in the text.

- Table 1: the color of the text under the microscopy pictures and the color of the highlighted boxes makes the figure confusing as it seems to imply a connection between the text and the boxes. Please use different colors or no colors.

We have used different colors now.

- Fig. 3B, why is the effect of anti-GM-CSF antibody less pronounced in the EVimDC compared to EVmaDC?

We do not know why the anti-GM CSF antibody did not block EV from imDC as efficient as from maDC. We could imagine that the density of GM CSF on the surface of EV is potentially different in vesicles from imDC compared to maDC. This question relates to the issue of how GM CSF is packaged into vesicles (see also comments by referee 1). We cannot give a satisfying answer to this question. Obviously more basic research has to be done.

In addition, the authors write in the results "Monocytes treated with imDC-EV, maDC-EV or recombinant GM-CSF revealed a strong Stat5 tyrosine phosphorylation as compared to non-stimulated cells. This

effect was blocked in the presence of an anti-GM-CSF antibody (Figure 3B, red box)." However, the mentioned figure does not show Stat5 phosphorylation but the % of FSChigh CD11b+ cells. The authors should fix the text or include the correct figure. The reduction of Stat5 phosphorylation could be also added to Figure 3A.

The referees correct. The description in the Figure panel is now corrected.

- Fig. 3C/D. There is a big difference in the levels of GM-CSF in EV imDC compared to EVmaDC in Fig. 3D, which can not be observed in Fig. 3B. Is there an explanation for this?

*I guess the referee is referring to Figure 3C/D and not B/D. Indeed, in the experiment in panel C the level of GM CSF is about the same in EV from imDC and maDC, whereas in Figure 3D, the difference is 3.6 fold (for this individual analysis). The discrepancy potentially reflects a variation in donors and assay systems. We have now added another example of a CCF analysis, where we saw the same difference in the protein array assay (**Figure S4C**). Also when we use the multiplex technology, we constantly recorded an increased vesicle-associated GM-CSF secretion as demonstrated in panel E. We have used both assay system to make sure that there is indeed an increase of factors in vesicles from maDC. After performing many of these assays, also for other projects, we think that the multiplex analysis system is more accurate and reliable.*

- Fig. 3D: "fold in. im/ma" should be the other way around (fold increase ma/im)

This has been corrected.

- All figures are labeled with Schierer et al... whereas tables are labeled as Ostalecki et al...

This is a mistake and has been corrected.

Reviewer #3 (Comments to the Authors (Required)):

This manuscript provides some intriguing data with regard to the potential mechanisms of immature DC differentiation by extracellular vesicles (EV). Although the functional phenotype looks interesting, the mechanistic support for the involvement of EV would need to be further solidified.

1). The key cytokine (or other soluble mediators) within the EV or released by EV responsible for the observed effect is not identified.

We think one of the key cytokines is GM CSF, as blocking of the cytokine abolishes the development of CD11b positive cells (Fig. 3B). On the other hand it is likely that several cytokines cooperate in the differentiation of monocytes.

2) To further build a compelling causative connection, approaches that can block the EV release should be applied to perform the functional studies.

This is certainly an excellent idea, but we feel this is beyond the focus of this paper. In this work here we first wanted to describe the phenomenon in principle.

October 29, 2018

RE: Life Science Alliance Manuscript #LSA-2018-00093-TR

Prof. Andreas Stephan Baur
University of Erlangen-Nürnberg
Department of Dermatology
Hartmannstrasse 14
Erlangen, Bavaria 91052
Germany

Dear Dr. Baur,

Thank you for submitting your revised manuscript entitled "Extracellular Vesicles from mature human DC differentiate Monocytes into immature DC" to Life Science Alliance.

We have now received input from original reviewer #1 and #2 on the revised version. As you know, we wanted to get strong support on the revised version from these reviewers in order to move forward here. However, and as you can see below, both reviewers are somewhat disappointed by the revision performed. While reviewer #2's first paragraph on the in vivo relevance and mechanism is not a concern for publication here, both reviewers are unhappy with the cytokine assays performed and think that their concerns have not been adequately addressed. Reviewer #2 currently does not support publication of your work at Life Science Alliance, while reviewer #1 is still somewhat supportive for publication here.

Given this input, I had asked you to provide a preliminary point-by-point response to the reviewers' concerns upfront. I appreciate your view on the criticisms raised, and I would like to invite you to provide a further revised version for publication here. This revised version should discuss the remaining concerns in a constructive way. The model figure can remain, but you may want to include and discuss alternative routes of EV secretion.

While revising your manuscript further (please also provide a final point-by-point response again), please pay attention to the following:

- please include 10 authors et al. in your reference list
- Table S1 is called S2 in the actual file, please fix
- in your ms text, you mention a Table 2, but there is no table 2
- FigS4D is called out in the legend but not in the ms and this panel doesn't exist in the figure
- please provide single-page files for each figure (in portrait format), figure 5 runs over two pages at the moment
- please make sure that all images have scale bars

A. FINAL FILES:

-- High-resolution figure, supplementary figure and video files uploaded as individual files: See our detailed guidelines for preparing your production-ready images, <http://life-science-alliance.org/authorguide>

B. MANUSCRIPT ORGANIZATION AND FORMATTING:

Full guidelines are available on our Instructions for Authors page, <http://life-science-alliance.org/authorguide>

Thank you for your attention to these final processing requirements.

Sincerely,

Reviewer #1 (Comments to the Authors (Required)):

The revised version of the article by Schierer et al answers partially my previous concerns, with a limited amount of new experiments.

The authors now show (as requested) quantifications of the amount of 25 cytokines in the EV pellets from maDCs used in the functional studies (figS2A, 26 columns, but not sure what is the cytokine measured in the first column: T MDPNCE?), and comparison EV/EV-free supernatant for 3 other cytokines (figS2B), which shows that at best EVs contain 40 times less cytokines than the amount released as free soluble form (for IFN γ) and over 1000 times less for IL10 and IL21. They do not show this quantification for EVs of imDC, which is a shame, since EVs from these 2 types of cells display different effects.

The question is of course now, why they did not measure the amount of the 25 cytokines of figS2A in EV vs EV-free supernatant (at least the ones with high detectable amounts, except maybe GM-CSF due to its strong presence as exogenous added cytokines). And also why they did not compare side-by-side the effects of EV-associated cytokines as compared to the same cytokines in a free soluble form (or at the amount detected in the EV-depleted supernatant)?

The authors only answer by giving the amount of cytokines (GMCSF, TNF α , IL6) used in classical in vitro differentiation protocols, but possibly lower concentrations of these cytokines (similar to those quantified in EVs) would, in their culture conditions, also induce DC differentiation! Thus the answer is not intellectually satisfying.

The other answer to my concern about the model is also unsatisfying: the authors base all their model on their previous work on another cytokine (TNF α instead of GM-CSF which they focus on here), in other cellular models (T cells, HEK cells, HIV-Nef protein...) than the work performed here. In addition, the authors explain themselves that GM-CSF is detected at the surface of their EVs by Flow cytometry (answer to my comment), whereas their model shows GM-CSF inside the secreted vesicles. Since their model contradicts all cell biology knowledge of intracellular trafficking, and their own data, and since it is not documented in any way here (for GM-CSF, in dendritic cells, etc), it must at least show the readers the 2 options : 1) classical EV secretion via MVBs or plasma membrane budding leading to outside-out oriented vesicles, with GM-CSF at the surface (possibly bound to its receptor or to other surface molecules), versus 2) their preferred model of release of packets of intracellular secretory vesicles with outside-in orientation followed by break-up of the enclosing membrane.

I don't want to prevent publication, if the authors can provide proper correction of the model figure (point 2), and at least justify their choice not to perform the suggested experiment and discuss better how endogenous EV secretion should be relevant in vivo when much more soluble cytokines

are also present (point 1).

Reviewer #2 (Comments to the Authors (Required)):

The authors have addressed some of my points. However, in my opinion the in vivo part is not convincing and does not demonstrate a direct role of EVs in monocyte differentiation. The mechanism and the biological relevance are missing.

The cytokine in the EV mediating the monocyte differentiation is not clear. They suggest it could be GM-CSF. However, GM-CSF was added to the culture conditions in order to generate DCs. Therefore, they would need to show that GM-CSF is indeed produced by DCs and contained in the vesicles and not a 'contamination' from the GM-CSF-containing medium. Could they use EVs derived from DCs generated with Flt3 ligand in the absence of GM-CSF to show that EVs indeed contain GM-CSF? Or as suggested previously, take EV-DCs from GM-CSF-deficient mice?

The revised version of the article by Schierer et al answers partially my previous concerns, with a limited amount of new experiments.

The authors now show (as requested) quantifications of the amount of 25 cytokines in the EV pellets from maDCs used in the functional studies (figS2A, 26 columns, but not sure what is the cytokine measured in the first column: TMDPNCE?),

Sorry, this is RANTES and the mislabeling happened during the conversion process from coral draw to pdf.

and comparison EV/EV-free supernatant for 3 other cytokines (figS2B), which shows that at best EVs contain 40 times less cytokines than the amount released as free soluble form (for IFN γ) and over 1000 times less for IL10 and IL21. They do not show this quantification for EVs of imDC, which is a shame, since EVs from these 2 types of cells display different effects.

We had the impression that the referee wanted to see examples of cytokine quantifications in order to get an impression on their relative quantity in EV. We choose EV from mature DC since they are the ones converting monocytes into immature DC.

The question is of course now, why they did not measure the amount of the 25 cytokines of figS2A in EV vs EV-free supernatant (at least the ones with high detectable amounts, except maybe GM-CSF due to its strong presence as exogenous added cytokines). And also why they did not compare side-by-side the effects of EV-associated cytokines as compared to the same cytokines in a free soluble form (or at the amount detected in the EV-depleted supernatant)? The authors only answer by giving the amount of cytokines (GM-CSF, TNF α , IL6) used in classical in vitro differentiation protocols, but possibly lower concentrations of these cytokines (similar to those quantified in EVs) would, in their culture conditions, also induce DC differentiation! Thus the answer is not intellectually satisfying.

I think these are all relevant questions, but slightly out of the focus of this paper. We wanted to show that extracellular vesicles can differentiate monocytes in vitro, and give at least a strong hint that this might also happen in vivo (as a definite proof is difficult to demonstrate). We are almost sure that the artificial in vitro stimulation of monocytes with high doses of cytokines does not occur in vivo, at least not in this fashion, and hence a detailed comparison of this stimulation with an EV-mediated stimulation might not produce meaningful results. Furthermore, we do not know all the constituents of DC-derived EV, which could include micro-RNAs and other factors not assessed and hence a side by side comparison, particularly if it doesn't work, would not provide a satisfying answer.

The other answer to my concern about the model is also unsatisfying: the authors base all their model on their previous work ...

I think it is fair to base a model on extensive previous work, even if it does not fully agree with the literature. For example, in our 2016 paper (Lee et al. 2013) we demonstrate that EV in HIV infection, containing most cytokine/chemokines including TNF and GM-CSF, do not stain for CD81, Tsg101 and CD25 (all typical exosome markers), and hence are likely not typical MVB-derived exosomes. That's why we never call them exosomes.

..... on another cytokine (TNF α instead of GM-CSF which they focus on here), in other cellular models (T cells, HEK cells, HIV-Nef protein...) than the work performed here. In addition, the authors explain

themselves that GM-CSF is detected at the surface of their EVs by Flow cytometry (answer to my comment), whereas their model shows GM-CSF inside the secreted vesicles. Since their model contradicts all cell biology knowledge of intracellular trafficking, and their own data, and since it is not documented in any way here (for GM-CSF, in dendritic cells, etc), it must at least show the readers the 2 options : 1) classical EV secretion via MVBs or plasma membrane budding leading to outside-out oriented vesicles, with GM-CSF at the surface (possibly bound to its receptor or to other surface molecules), versus 2) their preferred model of release of packets of intracellular secretory vesicles with outside-in orientation followed by break-up of the enclosing membrane.

For the revision we had modified our model (a variation of this model has already been published (Lee et al. 2013)) accommodating our own data and the comments of the referee, showing GM CSF on the surface of vesicles. Furthermore, we have changed the text in the discussion part accordingly, stressing that alternative explanations are possible, for example that all GM CSF is found on the surface of vesicles. Based on our own year-long experience, publications, extensive confocal imaging and electron microscopy, we believe that our model is in principle correct, even though it may not fully comply with the published literature. However, since this seems to be a contentious issue, we prefer to retract our model (Fig. 8B).

It is, however, disappointing that I cannot express my own view in a discussion model. After all, progress in science needs conflicting viewpoints. The referee points to the classical model of exosomes released through MVBs that fuse with the plasma membrane. But I have not seen a convincing electron micrograph demonstrating this mechanism. Notably, a recent review presents a good quality electron micrograph supposedly demonstrating this MVB release mechanism (James Edgar BMC Biology 2016 14:46). All I could see, however, was the same mechanisms that we have published in 2009 (see figure below left image), namely that at these EV traverses the plasma membrane seemingly through a ruptured or otherwise penetrated membrane area, and not through MVB membrane fusion or a budding process (see arrows in image below, right).

What are exosomes, exactly? BMC Biology 14(46) · June 2016

Figure 1: The left electron micrographs represents an unpublished image from my lab showing EV release from PMA stimulated monocytes. The middle image shows exosomes release from EBV-transformed B cells that was published recently.

I don't want to prevent publication, if the authors can provide proper correction of the model figure (point 2), and at least justify their choice not to perform the suggested experiment and discuss better how endogenous EV secretion should be relevant in vivo when much more soluble cytokines are also present (point 1).

In the discussion part we have provided an explanation why vesicles may be transferred directly into cells,

namely because it's more efficient with less off target effects. Another good argument for the site directed transfer of EV are findings by us and others in tissue (!) for mature TNF vesicles, also termed focal TNF in tissue, which are cited in the manuscript (Ostalecki et al., 2016 Figure 5; Yuan et al., 2017, eg. Fig. 8). It is very likely that cytokines like TNF and GM CSF are released in a similar fashion.

Reviewer #2:

The authors have addressed some of my points. However, in my opinion the in vivo part is not convincing and does not demonstrate a direct role of EVs in monocyte differentiation. The mechanism and the biological relevance are missing.

I would have to respectfully disagree with the referee. With respect to mechanism we demonstrate that EV are taken up by monocytes in vitro, and in vivo, and induce STAT3 signaling effects that are blocked by anti-GM CSF antibodies (in vitro). With respect to in vivo relevance, we demonstrate that only maDC-derived EV induces a lasting attraction of immune cells and beginning differentiation of monocytes. I realize that the referee wants to see experiments with knockout mice. However, I would like to argue that our unique imaging technology demonstrates tissue dynamics, strongly suggesting that DC-derived EV induce cellular differentiation processes.

The cytokine in the EV mediating the monocyte differentiation is not clear. They suggest it could be GM-CSF. However, GM-CSF was added to the culture conditions in order to generate DCs. Therefore, they would need to show that GM-CSF is indeed produced by DCs and contained in the vesicles and not a 'contamination' from the GM-CSF-containing medium. Could they use EVs derived from DCs generated with Flt3 ligand in the absence of GM-CSF to show that EVs indeed contain GM-CSF? Or as suggested previously, take EV-DCs from GM-CSF-deficient mice?

This is an important point raised by the referee and we have addressed this issue carefully in our original manuscript (Figure 3 C) demonstrating that only endogenously produced GM CSF is found in EV preparations and not recombinant GM CSF, which is used for monocytes stimulation. Supporting this finding, both monocyte-derived immature and mature DC are generated with recombinant GM CSF, but only maDC-derived EV have effects in vitro and in vivo. Both findings support our conclusion that no residual recombinant GM CSF was transferred to monocytes through EV.

November 19, 2018

RE: Life Science Alliance Manuscript #LSA-2018-00093-TRR

Prof. Andreas Stephan Baur
University of Erlangen-Nürnberg
Department of Dermatology
Hartmannstrasse 14
Erlangen, Bavaria 91052
Germany

Dear Dr. Baur,

Thank you for submitting your Research Article entitled "Extracellular Vesicles from mature human DC differentiate Monocytes into immature DC". We appreciate the introduced changes, and it is a pleasure to let you know that your manuscript is now accepted for publication in Life Science Alliance. Congratulations on this interesting work.

DISTRIBUTION OF MATERIALS:

Again, congratulations on a very nice paper. I hope you found the review process to be constructive and are pleased with how the manuscript was handled editorially. We look forward to future exciting submissions from your lab.

Sincerely,

Andrea Leibfried, PhD
Executive Editor
Life Science Alliance
Meyerhofstr. 1
69117 Heidelberg, Germany

t +49 6221 8891 502
e a.leibfried@life-science-alliance.org
www.life-science-alliance.org